# Case-control study of activities associated with SARS-CoV-2 infection in an adult unvaccinated population and overview of societal COVID-19 epidemic counter measures in Denmark

**Pernille Kold Munch**[1], **Laura Espenhain**[1], **Christian Holm Hansen**[1], **Tyra Grove Krause**[2], **Steen Ethelberg**[1,3]*

**1** Department of Infectious Disease Epidemiology and Prevention, Statens Serum Institut, Copenhagen, Denmark, **2** Division of Infectious Disease Preparedness, Statens Serum Institut, Copenhagen, Denmark, **3** Department of Public Health, Global Health Section, University of Copenhagen, Copenhagen, Denmark

* set@ssi.dk

## Abstract

Measures to restrict physical inter-personal contact in the community have been widely implemented during the COVID-19 pandemic. We studied determinants for infection with SARS-CoV-2 with the aim of informing future public health measures. We conducted a national matched case-control study among unvaccinated not previously infected adults aged 18–49 years. Cases were selected among those testing positive for SARS-CoV-2 by RT-PCR over a five-day period in June 2021. Controls were selected from the national population register and were individually matched on age, sex and municipality of residence. Cases and controls were interviewed via telephone about contact with other persons and exposures in the community. We determined matched odds ratios (mORs) and 95% confidence intervals (95%CIs) by conditional logistical regression with adjustment for household size and immigration status. For reference, we provide a timeline of non-pharmaceutical interventions in place in Denmark from February 2020 to March 2022. We included 500 cases and 529 controls. We found that having had contact with another individual with a known infection was the main determinant for SARS-CoV-2 infection: reporting close contact with an infected person who either had or did not have symptoms resulted in mORs of 20 (95%CI:9.8–39) and 8.5 (95%CI 4.5–16) respectively. Community exposures were generally not associated with disease; several exposures were negatively associated. Consumption of alcohol in restaurants or cafés, aOR = 2.3 (95%CI:1.3–4.2) and possibly attending fitness centers, mOR = 1.4 (95%CI:1.0–2.0) were weakly associated with SARS-CoV-2 infection. Apart from these two factors, no community activities were more common amongst cases under the community restrictions in place during the study. The strongest risk factor for transmission was contact to an infected person. Results were in agreement with findings of our similar study conducted six month earlier.

**Data Availability Statement:** This study was performed as a national disease surveillance project and for legal reasons data are not available

in the public domain nor in databases accessible for researchers. However, the interview data from this study may be made available in a de-identifiable format to other researchers upon reasonable request. For such, please contact Statens Serum Institut directly (email: serum@ssi.dk).

**Funding:** The author(s) received no specific funding for this work.

**Competing interests:** The authors have declared that no competing interests exist.

## Introduction

During the COVID-19 pandemic, most countries have made use of widespread restrictions affecting normal social life. The purpose has been to limit physical inter-personal interaction, in order to limit transmission of SARS-CoV-2. A wide range of measures, sometimes referred to as non-pharmaceutical interventions, have been implemented at varying time points throughout the pandemic [1]. In 2020 and 2021, a plethora of public health recommendations, and restrictions have been introduced and re-adjusted on a continuous basis. In Denmark, as in many other European countries, restrictions have involved public gatherings, the level of working from home, mandatory use of face masks, and regulations and lockdowns of restaurant/café, bar, nightclubs, sport activities, cultural events and more [2]. However, the impact these societal restrictions have had to reduce SARS-CoV-2 infection have rarely been subject to study and restrictions have commonly been introduced without a defined evidence base; potentially leading to mixed reactions in the populations they are applied to. This has been done for good reasons, obviously in a crisis situation, implementation of measures can often not await the results of scientific studies, nevertheless there is a need for more knowledge about how these regulations function in preventing SARS-CoV-2 infection under real life community settings.

By use of a case-control design, researchers all over the world have aimed to identify determinants, private and societal, for SARS-CoV-2 infection. Risk factors reported from previous studies include household overcrowding [3, 4], work in senior/health care [3, 5], work on-site [3, 4, 6, 7], foreign citizenship [3] and low education [3]. At the societal level, only few studies have been performed. When investigating activities such as use of public transport, frequenting restaurants/other dining spaces or bars, participating in indoor sports activities or buying food in stores, such studies have shown conflicting results [3–6, 8, 9].

Towards the end of 2020, we investigated societal activities associated with SARS-CoV-2 infection in Denmark by use of a case-control study design. This was done in a period where society was partially open with public gathering restrictions and mandatory face mask use indoors in public places, the original wild type (Index) strain of SARS-CoV-2 was the dominant strain in circulation and the COVID-19 vaccine rollout had not yet begun. We found that having had contact, in particular *close contact*, to another person with a known SARS-CoV-2 infection was strongly associated with infection. In contrast, only few community exposures were found to be associated with SARS-CoV-2 infection. They were participation in events where people sang, attending fitness centers and related to consumption of alcohol in bars. Other community exposures appeared not to be associated with infection e.g. supermarkets, public transport, and restaurants [10].

In June 2021, the COVID-19 situation had changed. The number of infected persons was declining, approximately 35% of the Danish population had received the first vaccination dose and approximately 20% the second. Society was gradually reopening, with now only societal restrictions in place for those individuals who were not vaccinated, or protected from previous infection and for those who did not have a recently negative SARS-CoV-2 test. Moreover, the Alpha SARS-CoV-2 variant was now circulating. In this situation, we again sought to identify societal activities associated with SARS-CoV-2 infection in Denmark. Here we present the results of a second national case-control study of risk factors for infection. For context and reference, we further created an overview of the official restrictions that have been in place in Denmark throughout the COVID-19 epidemic.

## Methods

### Officially imposed societal restrictions

We mapped public health measures and restrictions introduced in Denmark, in the period from February 2020 to March 2022. We covered measures within the following areas: public gathering restrictions (indoor, outdoor and at home); schools; workplaces; public spaces: grocery shops, non-essential shops, shopping malls, restaurants, bars/nightclubs, indoor cultural events, libraries, church/religious communities, public transport and sport activities. We categorized them into three different levels (open without restrictions, open with restrictions and fully locked down). The information was retrieved from relevant Danish government ministries and from the national COVID-19 communication partnership (coronasmitte.dk).

During the period of the case-control study, societal restrictions were mainly in place for those adults who were unvaccinated, had not previously recovered from SARS-CoV-2 infection or who had not recently tested negative for SARS-CoV-2. This status could be documented by use of a digital 'corona passport' (accessible via a smart phone app) first introduced in May 2021, or as a printable PDF with a QR code. The particular requirements for a valid corona passport within the study period was: I) Vaccination: from 14 days to 42 days after the first dose, or after the second dose (mRNA vaccines), or 14 days after dose one with Johnson & Johnson. After vaccination, the corona passport was valid for 8 months [11]. II) Negative official RT-PCR SARS-CoV-2 test: Taken within the past 72 hours [12]. III) Recovered after SARS-CoV-2 infection: Previously infected with COVID-19 documented by positive PCT test, performed at least 14 days and maximum 8 months prior [11].

### Case-control study design

To identify societal activities associated with SARS-CoV-2 infection in Denmark, at a fixed point in time during the pandemic, we conducted a national, individually matched, case-control study. The study methods were largely as those previously described [10], however with minor modifications. Eligible cases were unvaccinated individuals between 18–49 years old, not previously infected by 8 June 2021, with an address in Denmark, and an RT-PCR confirmed SARS-CoV-2 infection in the period from 8 to 12 June 2021. We listed eligible cases in random order and aimed to include the first 500 cases, who had not been hospitalized or traveled outside of Denmark during the exposure period. Controls were matched to cases by year of birth, sex (2 levels) and municipality (98 levels) and were extracted from the Danish Civil Registration System [13]. Only controls unvaccinated and not previously infected by 12 June 2021 were included to match the risk of the infection profile of the cases.

### Date sources

In Denmark, an extensive test system was built during 2020 and the first part of 2021. In addition to the clinical test system, RT-PCR tests were provided for all, without indication, through widely available, free-for-all public test stations. Information on SARS-CoV-2 tests was obtained at person-level format from the Danish Microbiology Database [14–16]. Controls were sampled from the Danish Civil Registration System from which information on age, sex, vital status, area of residence and country of birth was also obtained [17]. Information on vaccines administered against SARS-CoV-2 in Denmark are registered in the Danish Vaccination Registry [18]. Through this, person level information on vaccinations given, including the date of administration, was obtained. Information from other data sources were linked to by use of the unique civil registry number assigned to all Danish residents [17].

## Data collection

Cases and controls were interviewed via telephone between 15 June and 24 June, 2021, by a sub-contracted private polling institute. At least two attempts were made to call each eligible case and control per day. Controls were sought interviewed after their matched case had been interviewed. We aimed to include one matched control per case. We sampled 10 controls per case, but sampled an additional 10 controls in the instances where none of the first 10 had been reached within two days.

Compared to the study performed in November 2020, the period that our questions concerned was shortened from a 14-day to a 6-day period. The 6-day period ran from eight to two days prior to symptom onset (or test date if asymptomatic) for cases and the same 6-day period for their matched controls. We refer to this as the exposure period. Further, the questions related to contact exposures were updated due to changes in the national guideline set by the Danish Health Authority and therefore had a slightly different wording. Contact exposures included *close contact/other contact* with a person with known SARS-CoV-2 infection, with or without symptoms. The *close contact* definition was: Exposure to a household member, direct physical contact (for example hugging), unprotected and direct contact with secretions from an infected person, having been within a distance of less than 1 meter for more than 15 minutes, or caring for COVID-19 patients where the prescribed protective equipment had not been used. *Other contact* was defined as contact with a person with known SARS-CoV-2 infection. The community exposures inquired about were the same as in our first study, and included activities such as dining at restaurants, going to bars, shopping, participating in sport activities, and religious events or events involving singing etc. along with questions of if these activities took place indoors or outdoors, or involved consumption of alcohol. In contrast to the first study, we did not include questions on protective behavior and adherence with measures. For further information, please refer to [10].

The study was performed as a national disease surveillance project, registered with the Danish Data Protection Agency (reg no 21/04112) and specifically approved regarding legal, ethical and cyber-security issues. According to Danish legislation, approval from an ethical committee is not needed for medical studies not involving biological material.

## Statistical analyses and power calculation

The required sample size was calculated based on an expected bar visit frequency of 10% among controls [10]. With a power of 80%, an alpha-level of 0.05 and an odds ratio to detect at 2, we needed 566 participants following standard sample size formulae for unmatched case-control studies [19]. We assumed that 30% of all cases would be infected within the household and the required sample size was then calculated to be 810 (405 cases and 405 controls). We aimed to include a total of 1000 participants.

We compared basic demographic characteristics (country of origin, household size, number of contacts and employment status) of cases and controls. We used Chi-Square to test for overall differences and matched logistic regression to test for intergroup differences. We compared exposures reported by cases with those of controls using conditional logistic regression taking matching into account. For answers to secondary questions where matched analyses were not possible, logistic regression with adjustment for the matching variables was performed. We additionally adjusted for household size and migration background. For analyses concerning community exposures, we excluded cases (and their matched control) who reported to be infected in their household. As a sensitivity analysis, in order to assess self-isolation, we excluded cases and controls (and their respective matches) who reported having been close contact to an infected person during the exposure period. If such persons already during

the exposure period were aware that they were close contacts, they could have self-isolated, as recommended, or modified their behavior and thus would have been less likely to participate in activities in the community.

## Results

### Official COVID-19 counter measures in Denmark

The first public health measures were introduced in March 2020. During the following two years, a complex series of public health measures and restrictions were put in place, lifted and/ or reintroduced in response to the development of the epidemic. A detailed overview of these is given in Fig 1.

For the case-control study period in early June 2021, the following restrictions were imposed: Restaurants, cafes, bars etc. had to close at 11 pm with last servings at 10 pm. Use of face masks was mandatory for those aged 12 years or older, in indoor public spaces, including shops and public transport, except when seated at a table to eat or drink. Nightclubs were closed. At cultural, sport and religious events, a maximum of 500 seated people could gather, facing the same direction. Further, a valid corona passport was required for access to all public spaces, except pharmacies and shops selling foods. The maximum number of people allowed for spontaneous or private gatherings was 50 inside and 100 outdoor [20].

### Case-control study

In the inclusion period 1,565 unvaccinated adults aged 18–49 years, were diagnosed with SARS-CoV-2 and eligible for inclusion. A valid phone number was available for 1,148 (72%) cases, 829 were attempted contacted before 500 were included in the study and enrolment ended. A total of 529 matched controls were included in the study (Fig 2). Eligible and included cases were similar regarding age, sex and geographic region of residence. Compared to eligible cases diagnosed during the period, the recruited cases were less likely to have migrant background. Included cases and controls had similar household sizes, but the groups differed regarding migrant background and number of contacts (Table 1).

In total, 80% of cases reported knowing where they had been infected. This was primarily reported to have happened in the household (20%), at the workplace (16%), or among friends or family members (other than the household,16%). Cases further reported education facilities (5.4%), leisure activity (5.2%), other events (2.5%) or 'other place/exposure' (13%) as places of likely infection (Table 2). In total, 87% of the cases reported to have experienced symptoms of COVID-19.

Overall, 47% of the cases and 8% of the controls reported that they had been in contact with an infected person in the exposure period. Most reported 'close contact'. 'Other contact' with an infected person with symptoms was reported by 2.9% of the cases and 1.2% of the controls, resulting in a matched odds ratio (mOR) estimate of 3.3 (95% CI: 1.2–9.2). *Close contact* with an infected person without symptoms was reported by 15% of cases and 3.1% of controls had, resulting in a mOR of 8.5 (95% CI: 4.5–16). *Close contact* with an infected person with symptoms was reported by 27% of cases and 2.3% of controls, mOR: 20 (95% CI: 9.8–40, Table 3).

### Community determinants of SARS-CoV-2 infection

Controls were more likely to report having been to restaurants than cases, mOR: 0.66 (95% CI: 0.49–0.90). However, cases were more likely than controls to report, that they or others in their company had consumed alcohol during the restaurant visit, adjusted odds ratio (aOR): 2.3 (95% CI: 1.3–4.2). The same trend was seen for bar and indoor cultural events, where more

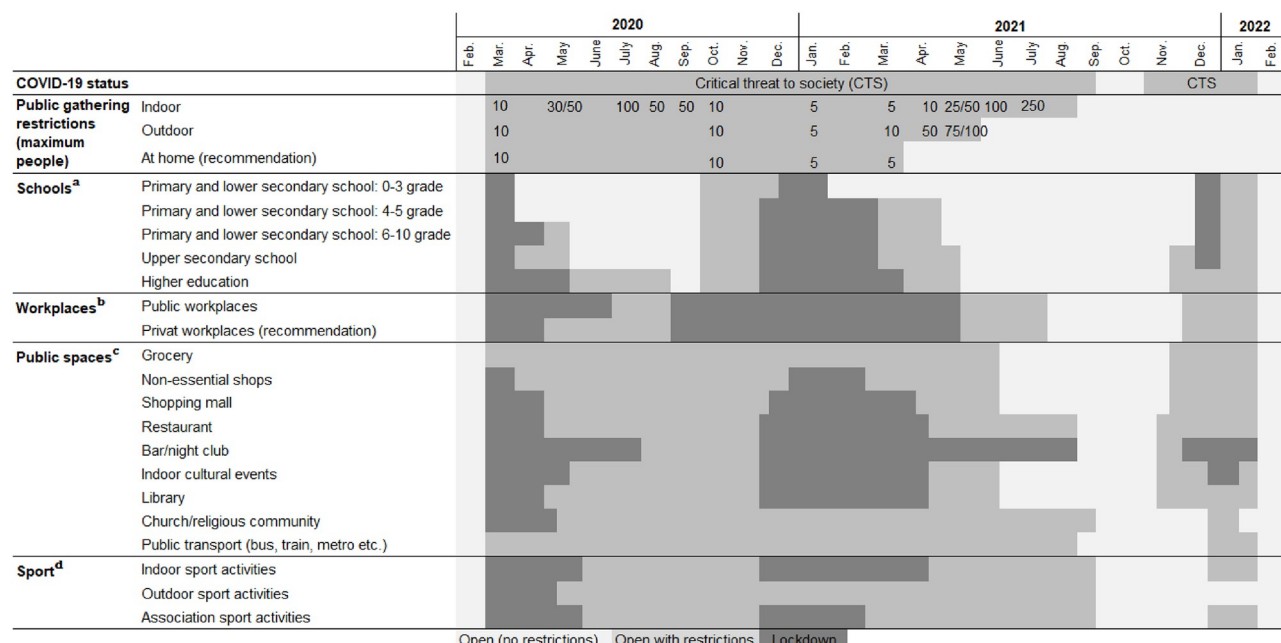

**Fig 1. Public health measures and restrictions (February 2020 to March 2022).** [a]: As of 16 March 2020, all schoolchildren and students were sent home. As of 15 April 2020, schoolchildren in grades 0–5 and final year students in upper secondary school returned to in–person learning. As of 18 May 2020, schoolchildren in grades 6–10 and all students in upper secondary school returned to in–person learning. As of 27 May 2020, institutions for higher education reopened for in–person learning when physical attendance were essential. As of 22 June 2020, academic institutions facilities for higher education students were allowed to open. As of 29 October 2020, primary school staff were allowed to wear face masks. Within upper secondary school and higher education, face masks were partially mandatory. As of 16 December 2020, all schoolchildren from grade 5, upper secondary and higher education underwent remote learning. Staff for grades 0–4 were allowed to wear face masks. As of 21 December 2021, schoolchildren and students not already on holiday were sent home. As of 8 February 2021, schoolchildren in grades 0–4 returned to in–person learning. As of 1 March 2021, final year students were allowed to return to 50% in–person learning in North–and West Jutland and all students at all education levels in Bornholm were allowed to return to 100% in–person learning. As of 15 March 2021, schoolchildren in grades 5–8 and graduating students who had not returned to 50% in–person learning in were allowed to attend outdoor classes once a week. Final year students were allowed to return to 50% in–person learning in Jutland, Funen, Vest and South Zealand and offshore islands. In offshore islands, primary and lower secondary school students returned 100% to in–person learning. For staff and students over 12 years of age, biweekly tests were recommended. For upper secondary schools and higher education, testing twice a week for students and staff was mandatory. As of 22 March 2021, students in the Capital Region were allowed to return to in–person learning in line with the rest of Denmark. As of 6 April 2021, schoolchildren in grades 5–8 were allowed to return to 50% in–person learning. Students within higher education with practical components to their studies were allowed return to 50% in–person learning and all other students were allowed return to 20% in–person learning. It was mandatory to get tested two times a week. As of 21 April 2021, 80% of learning was in–person for final year students and 30% for students in higher education (with the exception of the Capital Region) and there was the option for outdoor learning. Teaching for grades 5–8 was allowed to resume in–person if held outdoors during the weeks were in–person learning could not take place indoors. As of 6 May 2021, final year students and grades 5–8 returned to 100% in–person learning. Higher education in the Capital Region was allowed to take place in–person outdoors. As of 21 May 2021, all students were allowed to return to 100% learning in–person with the condition of getting tested twice a week. As of 29 November 2021, a valid coronapas was required within upper secondary and higher education. As of 15 December 2021, all schoolchildren in primary and lower secondary school underwent online learning. As of 19 December 2021, it was mandatory to wear face masks in upper secondary and higher education. As of 3 January 2022, it was recommended that those in upper secondary and higher education get tested twice a week. As of 5 January 2022, primary and lower secondary schools were allowed to return to 100% in–person learning, with the condition of students and staff getting tested twice a week. As of 1 February 2022, all restrictions were lifted. *Primary and lower secondary school*: 0–3 grade (schoolchildren age 5 to 10 years), 4–5 grade (schoolchildren age 9 to 13 years) 6–10 grade (schoolchildren in the age of 12 to 17 years). [b]: As of 13 March 2020, all employees were sent home. As of 14 April 2020, employees of private workplaces were allowed to return to work in–person. However, it was recommended that employees of private workplaces worked from home if possible. As of 27 May 2020, public workplaces in Jutland and Funen were allowed to return to office. As of 15 June 2020, public workplaces in Zealand were allowed to return to office. However, working from home and staggered working hours were recommended. As of 19 September 2020, employees of private and public workplaces were encouraged to work from home. As of 21 May 2021, 20% of the workforce from both public and private workplaces were allowed to return to office. As of 14 June 2021, 20% of the workforce from both public and private workplaces were allowed to return to office, with the recommendation of weekly testing. As of 1 August 2021, 100% of public and private* employees were allowed to return to office. As of 26 November 2021, valid coronapas was mandatory at public workplaces. As of 10 December 2021, employees of private and public workplaces were encouraged to work from home. As of 1 February 2022, all COVID–19 restrictions were lifted. *Recommendations, rather than restrictions, were provided for all private workplaces. For public workplaces, restrictions were provided for non–essential employees.* [c] *Shops*: As of 18 March 2020, shopping malls and non–essential shops were required to close. As of 11 May 2020, shopping malls were allowed to reopen with guidelines about space requirements. As of 29 October 2020, the use of face masks when shopping was mandatory, the sale of alcohol was prohibited after 10 PM, and shops larger than 2.000 $m^2$ had space requirements and supervisory guards. As of 17 December 2020, shopping malls and non–essentials shops larger than 5 000 $m^2$ had to close. As of 25 December 2020, all non–essential

shops were required to close. As of 1 March 2021, non–essential shops were allowed to reopen with restrictions (booking and space requirements). As of 13 April 2021, shopping malls smaller than 15.000 m2 were allowed to reopen. As of 21 April 2021, all shopping malls and non–essential shops were allowed to reopen. As of 14 June 2021, face masks were no longer mandatory in shopping malls or non–essential shops. As of 29 November 2021, use of face masks was mandatory for customers (not employees) when shopping. As of 19 December 2021, the use of face masks was mandatory for employees and valid coronapas and further space requirements were introduced. As of 1 February 2022, all COVID–19 restrictions were lifted. *Restaurants and bar/night clubs*: As of 18 March 2020, restaurants and bar/night clubs were required to close. As of 18 May 2020, restaurants and bars were allowed to reopen (restrictions on space requirements and opening hours). As of August 2020 opening hours for restaurants and bars were extended. As of 19 August 2020 opening hours were limited and the use of face masks was mandatory (except when seated). As of 16 December 2020, restaurants and bars were required to close. As of 21 April 2021, restaurants and bars were allowed to reopen (05 AM to 11 PM), with restrictions (booking, valid corona pas and space requirements). As of 11 June 2021, opening hours at restaurants were extended to 12 AM. As of 14 June 2021, no longer mandatory to use face masks indoor. As of 15 July 2021, opening hours were extended to 2 AM. As of 1 September 2021, all restrictions for restaurants were lifted and night clubs were allowed to reopen with valid coronapas. As of 12 November 2021, a valid coronapas was required at restaurants and bars. As of 10 December 2021, night clubs were required to close and restrictions on opening hours (12 AM) and mandatory use of face masks (only for customers) at restaurants were put in place. As of 19 December 2021, further restrictions were put in place regarding opening hours (11 PM) and mandatory use of face masks for employees and space requirements. As of 1 February 2022, all COVID–19 restrictions were lifted. *Public transport*: As of 16 March 2020, limited use of public transport was recommended and seat reservation was required (regional trains). As of 31 July 2020, the Danish Health Authority recommended face masks during rush hour. As of 22 August 2020, it was mandatory to use face masks on public transport. As of 14 June 2021, it was mandatory to use face masks while standing in public transport. As of 1 September 2021, it was no longer mandatory to use face masks in public transport. As of 29 November 2021, it was mandatory to use face masks on public transport. As of 19 December 2021, it was mandatory for passengers to have a valid coronapas on long–distance buses and trains. As of 1 February 2022, all COVID–19 restrictions were lifted. *Library, cultural activities, church and other religious communities*: As of 13 March 2020, the lockdown suspended all public cultural and religious activities. As of 18 May 2020, libraries were allowed to open for the loan of books and churches were allowed to reopen. As of 1 March 2021, outdoor cultural activities were allowed to resume. As of 22 March 2021, a maximum of 50 persons were allowed to gather at religious activities. As of 19 December 2021, restrictions on space requirements and face masks were implemented. As of 24 December 2021, a valid coronapas was required to attend public cultural activities. As of 16 January 2022, indoor cultural facilities were allowed to resume with mandatory use of face masks. No space requirements within the church and other religious communities. As of 1 February 2022, all COVID–19 restrictions were lifted. [d] As of 18 March 2020, the lockdown suspended all indoor and outdoor sport activities. As of 18 May 2020, outdoor sport activities were allowed to resume. As of 8 June 2020, indoor sport activities were allowed to resume. As of 26 October 2020, a maximum of 10 persons were allowed to gather. As of 16 December 2020, the lockdown suspended all indoor and outdoor sport activities. As of 1 March 2021, a maximum of 25 persons were allowed to gather outside for sport activities. As of 22 March 2021, a maximum of 50 persons were allowed to gather outside for sport activities. As of 21 May 2021, sport activities were allowed to resume with valid coronapas. As of 10 August 2021, all restrictions were lifted. As of 19 December 2021, mandatory to use face masks inside. Valid coronapas was required. As of 1 February 2022, all COVID–19 restrictions were lifted. *Travel restrictions*: During this period, travel restrictions were also in place. Some of the restrictions enacted by the Danish Ministry of Foreign Affairs advised against all non–essential travel worldwide, quarantine, coronapas etc. However, these restrictions were not mapped because the restrictions depended on the travel destination. *Coronapas*: As of 6 April 2021, coronapas was implemented as a part of the reopening of society. A valid coronapas was either: I) Completed primary vaccination. II) Negative official RT–PCR SARS–CoV–2 or antigen test: Taken within the past 72 hours. III) Recovered after SARS–CoV–2 infection: Previously infected with SARS–CoV–2 documented by positive test, performed at least 14 days and maximum 12 weeks prior. As of 21 April 2021, a valid coronapas was either: I) Completed primary vaccination. II) Negative official RT–PCR SARS–CoV–2 or antigen test: Taken within the past 72 hours. III) Recovered after SARS–CoV–2 infection: Previously infected with SARS–CoV–2 documented by positive PCR test, performed at least 14 days and maximum 6 months prior. As of 21 May 2021, a valid coronapas was either: I) Vaccination: from 14 days to 42 days after the first dose, or after the second dose (mRNA vaccines). After vaccination, the corona passport was valid for 7 months. II) Negative official RT–PCR SARS–CoV–2 or antigen test: Taken within the past 72 hours. III) Recovered after SARS–CoV–2 infection: Previously infected with SARS–CoV–2 documented by positive PCR test, performed at least 14 days and maximum 8 months prior. As of 28 May 2021, the Danish 'coronapas app' was introduced. As of 1 July 2021, a valid coronapas was either: I) Vaccination: from 14 days to 42 days after the first dose, or after the second dose (mRNA vaccines). After vaccination, the corona passport was valid for 7 months. II) Negative official RT–PCR SARS–CoV–2 test: Taken within the past 96 hours or negative official antigen test: Taken within the past 72 hours. III) Recovered after SARS–CoV–2 infection: Previously infected with SARS–CoV–2 documented by positive PCR test, performed at least 14 days and maximum 8 months prior. As of 7 July 2021, a valid coronapas was either: I) Vaccination: from 14 days to 42 days after the first dose, or after the second dose (mRNA vaccines). After vaccination, the corona passport was valid for 12 months. II) Negative official RT–PCR SARS–CoV–2 test: Taken within the past 96 hours or Negative official antigen test: Taken within the past 72 hours. III) Recovered after SARS–CoV–2 infection: Previously infected with SARS–CoV–2 documented by positive PCR test, performed at least 14 days and maximum 12 months prior. As of 29 November 2021, a valid coronapas was either: I) Vaccination: from 14 days to 42 days after the first dose, or after the second dose (mRNA vaccines). After vaccination, the corona passport was valid for 12 months. II) Negative official RT–PCR SARS–CoV–2 test: Taken within the past 72 hours or negative official antigen test: Taken within the past 48 hours. III) Recovered after SARS–CoV–2 infection: Previously infected with SARS–CoV–2 documented by positive PCR test, performed at least 14 days and maximum 12 months prior. As of 8 December 2021, a valid coronapas was either: I) Vaccination: from 14 days to 42 days after the first dose, or after the second dose (mRNA vaccines). After vaccination, the corona passport was valid for 7 months. After revaccination, the corona passport is valid. II) Negative official RT–PCR SARS–CoV–2 test: Taken within the past 72 hours or negative official antigen test: Taken within the past 48 hours. III) Recovered after SARS–CoV–2 infection: Previously infected with SARS–CoV–2 documented by positive PCR test, performed at least 14 days and maximum 12 months prior. As of 16 January 2022, a valid coronapas was either: I) Vaccination: from 14 days to 42 days after the first dose, or after the second dose (mRNA vaccines). After vaccination, the corona passport was valid for 5 months. After revaccination, the corona passport is valid. II) Negative official RT–PCR SARS–CoV–2 test: Taken within the past 72 hours or Negative official antigen test: Taken within the past 48 hours. III) Recovered after SARS–CoV–2 infection: Previously infected with SARS–CoV–2 documented by positive PCR test, performed at least 11 days and maximum 5 months prior.

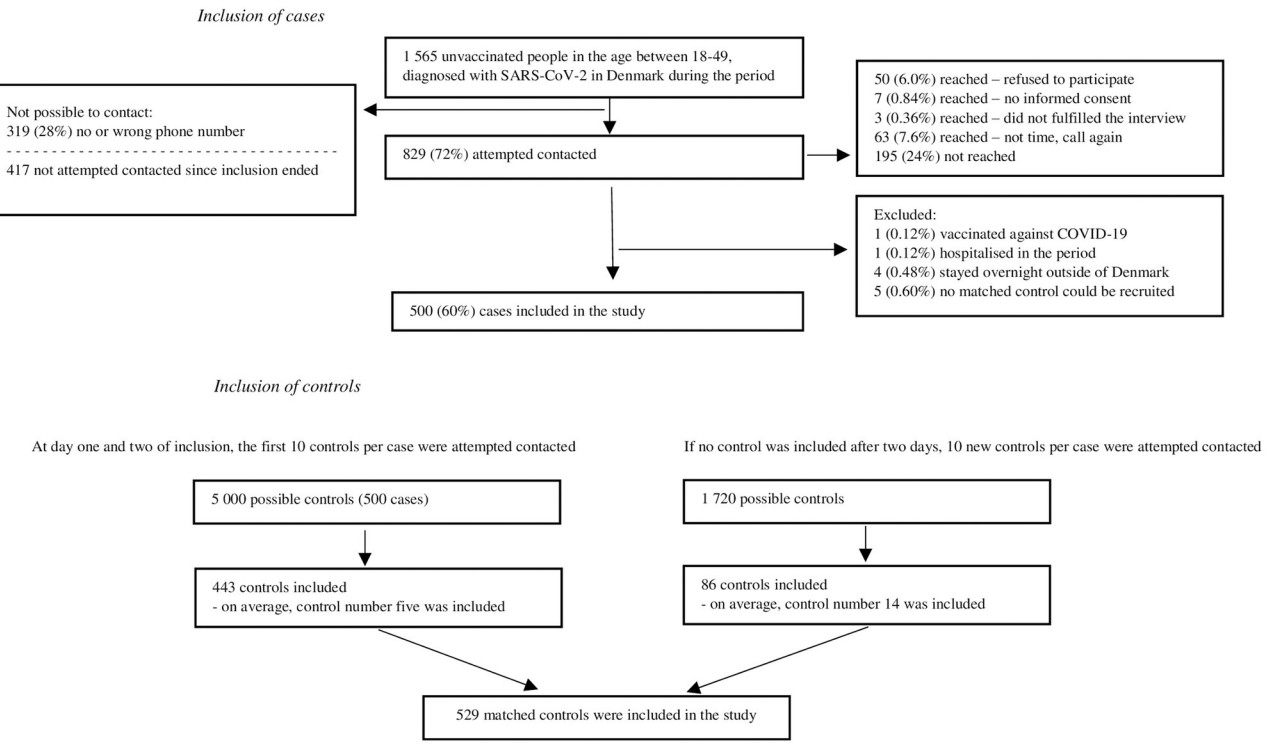

**Fig 2. Flow diagram illustrating inclusion of cases and control.**

controls than cases reported to have visited a bar and participated in indoor cultural events, but more cases than controls reported alcohol consumption (non-significant, Table 4).

In total, 24% of cases and 19% of controls had used a fitness centre at least once during the 6-day exposure period, mOR: 1.4 (95% CI: 0.98–2.00), this trend was not seen for indoor sport activities, mOR: 0.57 (95% CI: 0.32–1.0), nor for outdoor sport activities, mOR: 0.71 (95% CI: 0.49–1.0). For private social events, a higher proportion of controls than cases reported to have participated in small or medium-sized private social events, while there was no difference among the proportion participating in large private social events (Table 4). Finally, a higher proportion of controls than cases had visited shops/supermarkets, used public transport, or participated in religious events in the period. For participation in events, which involved singing, no difference was observed (Table 4).

In the sensitivity analysis, excluding cases and controls (and their matched case or control) who reported to have been close contacts to an infected person during the exposure period, we obtained largely similar results, showing the same trends (Table 5).

## Discussion

In this case-control study performed in June 2021, we found that known contact to an infected person was the most important risk factor for infection. For transmission in the community, outside the household, we identified only weak or no associations. Cases were more likely to have consumed alcohol while being on a restaurant or café and to have been at a fitness center than controls, the latter not being statistical significant. A large series of other societal activities were not found to be associated with SARS-CoV-2 infection; some were even found to be negatively associated with infection.

**Table 1. Number and proportion of included cases and controls by demographic characteristics and p–value of test for deviations, Denmark, June 2021.**

| Demographic characteristics | Included cases (n = 500), n (%) | Matched controls (n = 529), n (%) | P value[1] |
|---|---|---|---|
| *Age group* | | | na |
| 18–24 years | 199 (40) | 212 (40) | |
| 25–34 years | 159 (32) | 170 (32) | |
| 35–44 years | 85 (17) | 88 (17) | |
| 45–49 years | 57 (11) | 59 (11) | |
| *Sex* | | | na |
| Male | 263 (53) | 283 (54) | |
| Female | 237 (47) | 246 (47) | |
| *Region* | | | na |
| Capital Region of Denmark | 195 (39) | 200 (38) | |
| Region Zealand | 56 (11) | 62 (12) | |
| Region of Southern Denmark | 61 (12) | 66 (12) | |
| Central Denmark Region | 117 (23) | 128 (24) | |
| North Denmark Region | 71 (14) | 73 (14) | |
| *Migration background[2]* | | | *<0.05* |
| Denmark | 403 (81) | 462 (87) | Ref. |
| Western country | 26 (5.2) | 23 (4.3) | 0.34 |
| Non-western country | 71 (14) | 44 (8.3) | <0.05 |
| *Household size[3]* | | | *0.64* |
| 1 | 78 (16) | 91 (17) | Ref. |
| 2 | 141 (28) | 165 (31) | 0.92 |
| 3 | 109 (22) | 100 (19) | 0.22 |
| 4 | 105 (21) | 108 (20) | 0.46 |
| ≥5 | 67 (13) | 65 (12) | 0.39 |
| *Number of contacts* | | | *<0.05* |
| 0–5 | 290 (58) | 208 (39) | Ref. |
| 6–10 | 88 (18) | 156 (29) | <0.05 |
| 11–15 | 47 (9.4) | 51 (9.6) | 0.06 |
| 16–20 | 27 (5.4) | 30 (5.7) | 0.07 |
| 21–49 | 30 (6.0) | 52 (9.8) | <0.05 |
| Over 50 | 18 (3.6) | 32 (6.0) | <0.05 |
| *Employment status* | | | *0.34* |
| Employed | 307 (61) | 339 (64) | Ref. |
| Student | 150 (30) | 138 (26) | 0.06 |
| Other | 43 (8.6) | 52 (9.8) | 0.62 |

[1] P–value for Chi–Square–test for deviation between included cases and matched controls (italic). P–value for mOR (non–italic)

[2] Non–Danish migration background was defined as first or second generation immigrants from either Western countries (primarily neighboring European countries) or from non–Western countries (the five most frequent being Turkey, Iraq, Lebanon, Pakistan, and Somalia).

[3] Number of registered persons on the same address.

We prepared and present a detailed account of the officially imposed restrictions to free movement that were in place during the epidemic period from February 2020 to March 2022. Because of the possibility to perform population register–based studies and because of the particularly high number of SARS-CoV-2 tests that were performed in the country, Danish data has become a focus of interest in COVID-19 research [14, 16, 18, 21–27]. Besides its relevance for the current study, we hope with this account of the official restrictions imposed to be able

**Table 2. Number and proportion of likely place of infection as indicated by cases (n = 500)[1].**

| Likely place of infection | n (%) |
|---|---|
| Household | 105 (21) |
| Friends/other family than household | 80 (16) |
| Workplace | 84 (17) |
| Education | 28 (5.6) |
| Leisure activities | 27 (5.4) |
| Other events | 13 (2.6) |
| Public transport | 8 (1.6) |
| Other place/exposure | 68 (14) |
| Don't know | 103 (21) |

[1]It was possible to indicate more than one likely place of infection (516 answers included from 500 cases).

to provide context to studies of the epidemiology of COVID-19 being based on Danish SARS-CoV-2 data. For comparison of restrictive measures that were implemented in individual countries in the European Union, including Denmark, the European Centre for Disease Prevention and Control has published reports hereof [2].

**Table 3. Number, proportion and matched odds ratios (mOR) related to type of contact with a person with known SARS–CoV–2 infection (with/without symptoms), Denmark, June 2021.**

| Type of contact with infected person with/without symptoms | Cases (n = 483), n (%) | Controls (n = 512), n (%) | mOR (95% CI) |
|---|---|---|---|
| No contact with infected person | 256 (53) | 472 (92) | Ref. |
| *Other contact*, without symptoms | 10 (2.1) | 6 (1.2) | 3.04 (0.93–9.99) |
| *Other contact*, with symptoms | 14 (2.9) | 6 (1.2) | 3.25 (1.15–9.19) |
| *Close contact*, without symptoms | 73 (15) | 16 (3.1) | 8.53 (4.52–16.11) |
| *Close contact*, with symptoms | 130 (27) | 12 (2.3) | 20 (9.80–39.49) |
| *Migration background* | | | |
| Denmark | 391 (81) | 448 (88) | Ref. |
| Western country | 25 (5.2) | 23 (4.5) | 1.13 (0.52–2.43) |
| Non-western country | 67 (14) | 41 (8.0) | 2.17 (1.28–3.68) |
| *Household size* | | | |
| *1* | 75 (16) | 86 (17) | Ref. |
| *2* | 135 (28) | 159 (31) | 0.93 (0.56–1.54) |
| *3* | 105 (22) | 98 (19) | 1.07 (0.63–1.79) |
| *4* | 102 (21) | 106 (21) | 1.25 (0.72–2.18) |
| *≥5* | 66 (14) | 63 (12) | 0.88 (0.47–1.64) |

Note mOR estimates were adjusted for type of contact with/without symptoms, migration background and household size.

All 'do not know' responds were excluded in the analysis.

**Table 4. Number, proportion and odds ratios related to community exposures without household transmission, Denmark, June 2021.**

| Community exposures[1] | Cases (n = 395), n (%) | Controls (n = 421),n (%) | OR (95% CI) |
|---|---|---|---|
| *Restaurant or café* | 126 (32) | 176 (42) | 0.66 (0.49–0.90) |
| Alcohol vs. no alcohol | 35 (28) | 26 (15) | 2.33 (1.29–4.21) |
| *Bar* | 78 (20) | 102 (24) | 0.76 (0.53–1.09) |
| Alcohol vs. no alcohol | 65 (83) | 80 (78) | 1.31 (0.57–3.01) |
| *Indoor cultural events* | 23 (5.8) | 44 (10) | 0.58 (0.34–0.98) |
| Alcohol vs. no alcohol | 5 (22) | 7 (16) | 1.65 (0.34–7.87) |
| *Spectator at sport events* | 21 (5.3) | 33 (7.8) | 0.69 (0.39–1.22) |
| Alcohol vs. no alcohol | 6 (29) | 10 (30) | 0.59 (0.14–2.45) |
| *Indoor fitness center* | 95 (24) | 82 (19) | 1.40 (0.98–2.01) |
| *Indoor sport activities* | 24 (6.1) | 41 (9.7) | 0.57 (0.32–1.01) |
| *Outdoor sport activities* | 68 (17) | 94 (22) | 0.71 (0.49–1.03) |
| *Shopping (grocery)* | 299 (76) | 379 (90) | 0.36 (0.24–0.54) |
| *Shopping (other)* | 119 (30) | 205 (49) | 0.45 (0.33–0.61 |
| *Private social events <10 persons* | 125 (32) | 212 (50) | 0.46 (0.34–0.63) |
| Alcohol vs. no alcohol | 57 (46) | 93 (44) | 1.06 (0.65–1.72) |
| *Private social events 10–20 persons* | 48 (12) | 80 (19) | 0.61 (0.41–0.90) |
| Alcohol vs. no alcohol | 30 (63) | 49 (61) | 1.05 (0.45–2.46) |
| *Private social events >20 persons* | 34 (8.6) | 36 (8.5) | 1.05 (0.62–1.78) |
| Alcohol vs. no alcohol | 23 (68) | 25 (69) | 1.13 (0.32–4.03) |
| *Public transport* | 123 (31) | 179 (43) | 0.56 (0.41–0.78) |
| During rush hour | 43 (35) | 83 (46) | 0.59 (0.36–0.96) |
| *Religious events* | 5 (1.3) | 18 (4.3) | 0.29 (0.11–0.79) |
| *Events with singing* | 46 (12) | 51 (12) | 1.04 (0.65–1.65) |

For analyses on exposure mOR adjusted for migration background and household size are shown (italic). For sub–analyses on details within an exposure unmatched OR adjusted for sex, age, region, migration background and household size are shown (non italic).

[1]Never versus at least once in the period.

This study is the second in a series of two. Six month prior to the current study, we performed a first case-control study using similar methodology [10]. The main difference in set-up between the two studies related to the study size, the current was based on inclusion of 1000 cases and controls, the first study on 600 only, and the fact that the exposure period inquired about was shortened from two weeks to six days, with the aim of increasing the specificity. Apart from that, the main changes concerned external factors: the differences in restrictions in place, society this time being far more open, the Alpha rather than the original wild type SARS-CoV-2 viral strain being dominant and, importantly, the older population segments and other risk groups having been vaccinated and therefore excluded from the study population. The pattern of risk factors seen in the current study was remarkably similar to what we found in our previous study, where also, apart from contact to infected persons, fitness centers and alcohol consumption in bars and in addition participation in events which involved singing, were identified as being associated with SARS-CoV-2 infection. This second study may therefore be seen as corroborating the findings of the first and it would appear that besides direct contact with infected individuals, under the restrictive measures in place in Denmark, social activity involving alcohol and possibly the use of fitness centers constituted actual

**Table 5. Number, proportion and odds ratios related to community exposures without close contact cases and controls, Denmark, June 2021.**

| Community exposures (n cases/n controls)[1] | Cases (n = 267), n (%) | Controls (n = 277), n (%) | OR (95% CI) |
|---|---|---|---|
| *Restaurant or café* | 78 (29) | 108 (39) | 0.59 (0.39–0.89) |
| Alcohol vs. no alcohol | 17 (22) | 14 (13) | 1.72 (0.75–3.94) |
| *Bar* | 55 (21) | 72 (26) | 0.78 (0.50–1.22) |
| Alcohol vs. no alcohol | 44 (80) | 56 (78) | 1.00 (0.38–2.68) |
| *Indoor cultural events* | 12 (4.5) | 28 (10) | 0.48 (0.24–0.96) |
| Alcohol vs. no alcohol | 2 (17) | 8 (29) | 1.25 (0.04–41.13) |
| *Spectator at sport events* | 14 (5.2) | 23 (8.3) | 0.55 (0.27–1.10) |
| Alcohol vs. no alcohol | 4 (29) | 6 (26) | 0.36 (0.03–4.50) |
| *Indoor fitness center* | 61 (23) | 51 (18) | 1.58 (0.92–2.37) |
| *Indoor sport activities* | 19 (7.1) | 26 (9.4) | 0.84 (0.42–1.66) |
| *Outdoor sport activities* | 44 (16) | 59 (21) | 0.74 (0.46–1.20) |
| *Shopping (grocery)* | 202 (76) | 253 (91) | 0.36 (0.22–0.60) |
| *Shopping (other)* | 80 (30) | 142 (51) | 0.42 (0.29–0.63) |
| *Private social events <10 persons* | 85 (32) | 148 (53) | 0.40 (0.27–0.59) |
| Alcohol vs. no alcohol | 34 (40) | 74 (50) | 0.74 (0.40–1.35) |
| *Private social events 10–20 persons* | 30 (11) | 54 (19) | 0.53 (0.32–0.87) |
| Alcohol vs. no alcohol | 15 (50) | 33 (61) | 0.40 (0.12–1.30) |
| *Private social events >20 persons* | 15 (5.6) | 23 (8.3) | 0.64 (0.31–1.33) |
| Alcohol vs. no alcohol | 10 (67) | 16 (70) | 0.65 (0.07–5.64) |
| *Public transport* | 84 (31) | 118 (43) | 0.57 (0.38–0.86) |
| During rush hour | 29 (35) | 58 (49) | 0.48 (0.26–0.90) |
| *Religious events* | 1 (0.37) | 8 (2.9) | 0.13 (0.02–1.07) |
| *Events with singing* | 25 (9.4) | 35 (13) | 0.70 (0.38–1.30) |

For analyses on exposure mOR adjusted for migration background and household size are shown (italic). For sub–analyses on details within an exposure unmatched OR adjusted for sex, age, region, migration background and household size are shown (non italic).

[1]Never versus at least once in the period.

risk factors. As also speculated previously [10], activities potentially involving heavy breathing, excretions of aerosols, close interaction and touching of multiple surfaces in a closed indoor environment could constitute an environment prone to SARS-CoV-2 transmission, compared to outdoor sport activities. In the same line is the association with alcohol when present at restaurants or cafés, where alcohol may be a proxy for reduced awareness of protective behavior and adherence to IPC recommendations, increasing the likelihood of closer interactions and with that SARS-CoV-2 transmission. On the other hand, many other investigated activities, notably use of public transportation, supermarkets and cultural and sports gatherings were as common in cases than amongst controls.

Several other case-control studies of community determinants have been published. Early in the pandemic (May to June 2020), a case-control study conducted in Ohio and Florida, found no association between infection with SARS-CoV-2 and attending private or public gatherings or use of public transport [6]. Another study from July 2020 in the USA among outpatients also did not indicate an association between SARS-CoV-2 infection and use of public transport, shopping or visiting friends and family. However, cases were more likely to have been dining at restaurants and visiting bars/coffee shops than controls [9]. In Portugal, in September to October 2020, no association was found with use of public transport, restaurant visits, mall/supermarket visits, attending gym or sports activities and being infected with

SARS-CoV-2 [3]. A large case-control study from France (October and November 2020), in a period with broad-reaching public health and social measures, found an increased risk of infection associated with bar and restaurant visits, but no association with attending cultural gatherings [4]. Later in the pandemic, in a period with Delta circulation in France, another case-control study was performed (May to August 2021). Here people under 40 years of age attending bars, nightclubs or private parties were found to be at increased risk of infection. For public transport, cases were more likely to have used the subway, but not buses, trams or trains. For private gatherings, there was an association with ceremonies, but no association with other private gatherings, cultural events nor shopping (except from convenience stores). No association was seen for outdoor sports activities, but for indoor sport activities [8]. Another Danish case-control study, performed in October to December 2020 found associations similar to those seen in our studies [5]. The most important risk factor identified was contact to an infected person. Moreover, this study also found an association with fitness centers but not with shopping, use of public transport and participating in outdoor sport activities. Contrary to our findings, participation in indoor sport activities, larger events, and restaurant and bar visits were identified as risk factors [5].

Taken together, the available literature has not been able to show an association between SARS-CoV-2 infection or hospital admission and potential risk factors such as: supermarkets, outdoor sport activities or use of public transport in situations where basic preventive measures–mask use, keeping a distance–have been in place. Certain other activities have been found to be associated with infection, such as indoor sport activities or restaurant and bar visits. However, this appeared to vary, depending on the setting of the particular study.

Methodological strengths and limitation outlined in our previous 2020-study also apply for the current study. Among the limitations of the first study was the small sample size, therefore we went from 600 to 1000 participants to strengthen the power of the present study. We did not find any high mOR with wide confidence intervals, which could indicate risk of type II errors, and therefore we consider our sample size to have been sufficient to identify associations had they existed. Compared to our first study, we also shortened the exposure period inquired about, aiming to provide more specific estimates of associations. The use of the Danish Vaccination Registry enabled us to swiftly and objectively exclude those who had been vaccinated by the time of the study and used the Danish Microbiology Database to exclude those previously infected.

A potential bias would arise from systematic differences in behavior between cases and controls. Persons who recently had been in close contact with a person with SARS-CoV-2 infection, would, if they were aware of the exposure at the time, likely have been in self-isolation and would therefore not be active in the community. Because we for many activities found controls to be more exposed than cases (resulting in OR estimates below 1), we were suspicious of such a bias being at play. To explore this further, we performed a sensitivity analysis, in which we excluded all participants who reported to have been close contacts to infected persons. This did not change the results. We were able to exclude cases exposed in their household, which allowed us to produce a more specific estimate to risk associated to activities in the community. Unfortunately we did not have detailed enough information about the setting of the close contacts not part of the household, to include only those occurring during activities in the community. This means that we may have included cases which may have been infected at home or during non-community activities, which may have weakened the associations. Another potential systematic difference between cases and controls is their ability to recall activities in the period in question, if cases already during contact tracing had been asked to recall relevant activities, they may have been more likely to report such activities adequately, which could have influenced the results. Another potential concern relates to the

selection of controls. We used matched controls sampled from the general population, which was made possible because of our access to the Danish Civil Registration System. A different approach, which we did not opt for, would have been control selection with recruitment from the pool of persons testing negative in PCR test in the same period as the pool of cases tested positive. This option has been used by others, and we cannot say how it would have influenced on our results [5, 9]. In the study period, participation in societal activities generally required regular testing regardless of symptoms as part of the corona passport strategy in our unvaccinated, not previously infected study population. Persons who had had contact with a SARS-CoV-2 infected person or had symptoms of COVID-19 were recommended one or more tests and additionally recommendations or regular screening tests existed for certain professions [28, 29]. During the period around 11,000–26,000 RT-PCR tests were performed per 100,000 population [30] and seroprevalence studies based on blood donors showed that a low proportion the adult healthy population of Denmark had been infected [31]. Therefore, we believe that controls were unlikely to have been positive without knowing.

In conclusion, we show results of a study of risk factors for SARS-CoV-2 infection and compare with a similar study done six month earlier. Under the constraints of the methodology of case-control studies, no major community determinants for infection were identified with the exception of alcohol consumption and possibly use of fitness centers. We conclude that transmission in the general community was of little importance, while instead, the major risk factor for transmission was contact to a known infected person and that transmission primarily took place via infected colleagues or family members. Our study could not directly measure the effect of the societal restrictions in place but it is not unreasonable to expect that these had an effect in reducing any potential risks associated with community activities such as participating in cultural events, dining at restaurants, shopping and public transportation. Finally, we provide a timeline of non-pharmaceutical interventions that were implemented in Denmark from February 2020 to March 2022.

## Acknowledgments

We thank the participants of this study for taking their time to answer questions. We thank Caroline Eves for critical reading of the manuscript.

## Author Contributions

**Conceptualization:** Laura Espenhain, Christian Holm Hansen, Tyra Grove Krause, Steen Ethelberg.

**Data curation:** Pernille Kold Munch, Laura Espenhain, Christian Holm Hansen.

**Formal analysis:** Pernille Kold Munch, Laura Espenhain, Christian Holm Hansen.

**Funding acquisition:** Pernille Kold Munch, Tyra Grove Krause, Steen Ethelberg.

**Investigation:** Pernille Kold Munch, Laura Espenhain.

**Methodology:** Pernille Kold Munch, Laura Espenhain, Tyra Grove Krause, Steen Ethelberg.

**Project administration:** Pernille Kold Munch, Steen Ethelberg.

**Resources:** Tyra Grove Krause, Steen Ethelberg.

**Supervision:** Steen Ethelberg.

**Validation:** Pernille Kold Munch.

**Visualization:** Pernille Kold Munch.

**Writing – original draft:** Pernille Kold Munch.

**Writing – review & editing:** Laura Espenhain, Christian Holm Hansen, Tyra Grove Krause, Steen Ethelberg.

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
