## [Decision Letter · Decision Letter 0]

17 Jun 2022

PONE-D-22-13314Societal COVID-19 epidemic counter measures and activities associated with SARS-CoV-2 infection in an adult unvaccinated population – a case-control study in Denmark, June 2021PLOS ONE

Dear Dr. Ethelberg,

Thank you for submitting your manuscript to PLOS ONE. After careful consideration, we feel that it has merit but does not fully meet PLOS ONE’s publication criteria as it currently stands. Therefore, we invite you to submit a revised version of the manuscript that addresses the points raised during the review process.

ACADEMIC EDITOR:The reviewers agree that the study is interesting but made many constructive suggestions to improve it.Please address all of these before resubmitting.

We look forward to receiving your revised manuscript.

Kind regards,

Joël Mossong, PhD

Academic Editor

PLOS ONE

Journal Requirements:

Reviewers' comments:

Reviewer's Responses to Questions

**Comments to the Author**

1. Is the manuscript technically sound, and do the data support the conclusions?

Reviewer #1: Partly

Reviewer #2: Yes

Reviewer #3: Partly

Reviewer #4: Partly

2. Has the statistical analysis been performed appropriately and rigorously? 

Reviewer #1: Yes

Reviewer #2: Yes

Reviewer #3: Yes

Reviewer #4: No

3. Have the authors made all data underlying the findings in their manuscript fully available?

Reviewer #1: Yes

Reviewer #2: Yes

Reviewer #3: Yes

Reviewer #4: Yes

4. Is the manuscript presented in an intelligible fashion and written in standard English?

Reviewer #1: Yes

Reviewer #2: Yes

Reviewer #3: Yes

Reviewer #4: Yes

5. Review Comments to the Author

Reviewer #1: The study is a case-control study done in Denmark to elucidate factors associated with SARS-CoV-2 infection. Findings from this paper may be context-specific (in the setting of the situation in Denmark, in the setting of the alpha-dominant period, etc.). Still, the study is methodologically sound and provides important insights into controlling SARS-CoV-2. There are several points the authors should address. A couple of other minor points suggested to improve the paper further are also included.

1. The author’s conclusion in the abstract and the main text should be modified so that it is directly supported by the data. Specifically, the following statement is not supported directly by the data: “Transmission of disease through involvement in community activities appeared to occur only rarely, suggesting that community restrictions in place were efficient.” According to Table 2, at least based on the self-reported place of infection, over half of the cases are infected in the community activities. Also, even if the community restrictions in place were efficient, behaviors that are truly high-risk would have been identified as such in the present study design.

2. In the abstract, “confined space” and “contact to known persons” should be separated. “Contact to known persons” should be reworded to “contact to infected individuals.”

3. In the abstract, the sentence “For reference, we provide a timeline of non-pharmaceutical interventions in place in Denmark from February 2020 to March 2022” should be in the earlier part of the abstract to flow better.

4. In the abstract, the citation should not be included and should be removed.

5. If there are any missing data, they should be elaborated on in the tables and provide information on how they were addressed. If not, please ignore this comment.

6. This is a suggestion, but authors should reconsider whether there is a need to do the analysis excluding cases who reported to be infected in their household (Table 4) when the authors are doing an analysis excluding individuals with close contact (Table 5)? I understand that the former data provided some factors that are statistically significant while the latter did not, but it seems arbitrary.

7. The reason to exclude cases who are close contact is partly that cases most likely acquired the virus through this specific contact rather than other community exposures you are asking about in the questionnaire. This can also be mentioned in the manuscript.

Reviewer #2: The manuscript "Societal COVID-19 epidemic counter measures and activities associated with SARSCoV-2 infection in an adult unvaccinated population – a case-control study in Denmark, June 2021" is a valuable contribution. Although it does not providing ground-breaking or surprising results, it adds evidence to a steep learning curve from COVID-19 pandemic for epidemiologists, social scientists and policymakers. This contribution is a good example of robust and rapid evidence for policy that can be immediately applied during the crisis and build evidence to better prepare for future pandemics.

General suggestions on methods:

- Since the outcome of interest is tested positive for SARS-CoV-2, could the authors elaborate on the national criteria for testing that were used during the study period? It is relevant for the international audience, since in every country a "case" could have a slightly different probability of being identified. It is mentioned in the discussion that there was massive testing, possibly including asymptomatic people without any indications, in the occupational setting. In such a case, it is worth introducing this context, and addressing the issue of RT-PCR validity and reliability in the limitation section.

- Could the authors explain a bit more why they applied individual matching? With such massive exposures and such common infection, it would be perhaps better to use unmatched design and check for the effect of age and sex on the infection risk? The purpose of the control group is to represent exposures in the source population and matching is a way to prevent confounding. If age and sex are not confounders in a given scenario, matching can actually introduce confounding. I recommend adding to the Methods section a short explanation/rationale to apply individual matching.

- Could the authors explain the rationale behind excluding residents previously vaccinated and previously tested positive by RT-PCR? We know now that previous infection or vaccination does not prevent new infections. If one wants to investigate the effect of community exposures on disease risk, maybe it would be a better idea to not exclude infections occuring >6 months previously, for example. Maybe the authors just assumed that with a new disease circulating for a short time, it is safe to assume that all were recently vaccinated and all infected to date were "recently" infected and thus were immune to SARS-CoV-2 infections... These assumptions should be however better explained and addressed in the limitations section.

- I recommend using the term matched odds ratios (mOR) instead of odds ratios, to reflect the methodological approach. It should be updated in the entire manuscript, including the abstract.

- As mentioned in previous comments, the study has several limitations, not thoroughly revised by the authors. I recommend to be a bit more comprehensive in this matter!

Minor corrections:

- Abstract line 10: Please move the numbers to the results

- Abstract lines 20-21: Do not include citations in the abstract.

- Introduction line 39: Why to refer to case-control studies? Do authors refer to studies in Denmark? It would be good to be more precise...

- Methods line 70: typo in "ministries"

- Results line 163: did the authors mean: "Compared to eligible cases, eligible controls [...]"?

- Discussion 279: "conducting" not "conduction"

Reviewer #3: This is a case-control study attempting to identify individual and contextual risk factors for SARS-CoV-2 infection in Denmark. The study is well written and has a sound methodology. However, I have a few questions that I believe should be addressed:

- In the abstract (line 4) the authors mention they aim at testing the efficiency of such measures - however efficiency requires a resources dimension - do the authors believe they have addressed this?

- At the end of introduction (lines 60-61) authors mention the "present an overview of the official restrictions". This seems a bit misleading as the authors actually prepare one as part of their work. I suggest this to be rephrased.

- Authors have excluded hospitalized and vaccinated patients. Can they comment on how this might have influenced the results and how this can affect results generalizability?

- Data collection: The exposure period ranged from eight to two days before the symptom onset (or test for asymptomatic) for cases. However I could not identify what was the index date for controls - what was the 6-day refer to? Can the authors clarify?

- Some of the analysis reported in Table 1 and not described in the methods. I suggest this to be added.

- The authors emphasize the alcohol consumption in their results. However this has not been given an in-depth consideration in the discussion. Why the authors believe they have identified this results? Is it really related with the alcohol consumption or it this a proxy for other types of behaviours?

- There are negative results in for a series of contexts. Have the authors explored these results in more depth? They raise an hypothesis is the discussion (lines 261 onwards) which has not been corroborated by the sensitivity analysis. Do you see any alternative reasons for these results. Could remote work play a role here?

- The authors claim their results support the "efficiency" of existing measures. However I am not entirely convinced here. In fact there is no counterfactual to assess the effectiveness (which I believe is what in fact authors intended to assess) or the impact of those measures. Please comment.

Reviewer #4: The authors aim to identify determinants of Sars-COV2 infection in the Denmark in June 2021 using a case –control study. They compare exposure to determinants among

1) 500 cases unvaccinated; 18-49 years old, residents in Denmark, with a RT-PCR test positive result between 8-12 June that were not hospitalized; had not travelled abroad

2) With 529 controls matched on year of birth, sex and municipality with the cases (1 to 1 match).

The 6 days (from Day -8 to Day -2 prior to symptom or test +) exposure to determinants was measures by telephone interviews between 15 and 24 June. Conditional logistic regression is correctly use for the analysis.

The population from which cases and controls are coming from does not look identical concerning previous infection which is an exclusion criteria for controls but no clear info is provided about the cases. For the community transmission, given the restrictions in place, it means that we could have only controls with a recent negative test but we could have cases with a previous infection or a recent negative test who access public spaces. A subgroup analysis could be performed excluding cases with previous infection and their matched controls, to account for that.

In order to understand the results without the need to consult other articles a more detailed description of the questionnaire used is necessary.

The conclusion that Contact with another individual with a known infection as the main determinant for SARS-CoV-2 infection is based on sound methodology and nicely reflected in the different sections. Given the strength of the association even adjusting for other determinants will not modify the findings and or conclusions.

Sub-group analysis for the community transmission section.

The analysis for each determinant is performed separately adjusted for migration back ground, household size and the matching variables. There is no analysis performed by adjusting for all determinants, or for determinants associated with risk of infection in the bivariate analysis.

Based on the results from table 4 some of the interpretations about differences between cases and controls are not fully correct. In the results section it states that cases are more likely than controls to have consumed alcohol in a bar. By looking at table 4, the CI95% of the OR for alcohol consumption in a Bar includes the 1 meaning that the difference is not statistically significant. The same logic applies for the indoor sport activities, indoor fitness center and outdoor sport activities. The difference between cases and controls is not statistically significant. The results and discussion sections are to be updated accordingly concerning the community transmission part. (no significant difference for indoor fitness centers, indoor sport activities, alcohol consumption in bars )

It seems that the study is underpowered to detect community determinants of SARS-COV2 infection. Considering a matched 2 controls 1 case design could partially resolve that problem. In the multivariate model, only determinants associated with outcome in the bivariate analysis could be entered.

Other

Page 4 line 56 (Introduction section) Please rephrase

Society was gradually reopening, with now only societal restrictions in place for those individuals who were vaccinated, had recovered from infection or recently tested negative

In contradiction with Line 71-72 (methods section)

Table 1 Please provide OR for non matching variables. From the bivariate analysis it looks like the migration background and the number of contacts are associated with the risk of becoming a case.

In table 2 there is a total of 516 cases reported whereas in the results section only 500 cases are reported as included, please double-check or clarify

Table 3 Please provide adjusted OR for migration background and household size as well.

General Feedback

Please standardize the reporting of OR and CI95% by using everywhere 2 decimals (1,01 instead of 1,0 or 39,00 instead of 39).

Inclusive tables with absolute numbers and proportions as well as OR(CI95%) of both bivariate and multivariate analysis would be more easy to read and interpret.

6. PLOS authors have the option to publish the peer review history of their article (what does this mean?). If published, this will include your full peer review and any attached files.

Reviewer #1: No

Reviewer #2: **Yes: **Pawel Stefanoff

Reviewer #3: **Yes: **Andreia Leite

Reviewer #4: No

---

## [Author Response · Author response to Decision Letter 0]

7 Aug 2022

Reviewer #1: The study is a case-control study done in Denmark to elucidate factors associated with SARS-CoV-2 infection. Findings from this paper may be context-specific (in the setting of the situation in Denmark, in the setting of the alpha-dominant period, etc.). Still, the study is methodologically sound and provides important insights into controlling SARS-CoV-2. There are several points the authors should address. A couple of other minor points suggested to improve the paper further are also included.

1. The author’s conclusion in the abstract and the main text should be modified so that it is directly supported by the data. Specifically, the following statement is not supported directly by the data: “Transmission of disease through involvement in community activities appeared to occur only rarely, suggesting that community restrictions in place were efficient.” According to Table 2, at least based on the self-reported place of infection, over half of the cases are infected in the community activities. Also, even if the community restrictions in place were efficient, behaviors that are truly high-risk would have been identified as such in the present study design.

Answer: Thank you for this comment. The abstract and the conclusion in the main text have been modified, so the conclusion is supported by our results. We have included that, as we found no single determination to be associated with SARS-CoV-2 infection and because transmission appeared to primarily take place via contact to a person with known SARS-CoV-2 infection, community restrictions in place were adequate to reduce risk to a level where we could not find a difference between cases and controls.

Abstract (Lines 20-23): Alcohol consumption and fitness centers were weakly associated with infection, in agreement with findings of our similar study conducted six month earlier. Apart from the two factors, no community activities were more common amongst cases, suggesting that community restrictions in place were adequate. The strongest risk factor for transmission was contact to an infected person. 

Conclusion (Line 296-304): In conclusion, we show results of a study of risk factors for SARS-CoV-2 infection and compare with a similar study done six month earlier. Under the constraints of the methodology of case-control studies, no major determinants for infection were identified. Since no determinants for infection were identified, we conclude that overall societal restrictions in use in the spring of 2021 were adequate to reducing any potential risk associated with community activities such as participating in cultural events, dining at restaurants, shopping and public transportation; exceptions being fitness centers and alcohol consumption. Instead, a major risk factor for transmission was contact to an infected person and transmission primarily took place via infected colleagues or family members. Finally, we provide a timeline of non-pharmaceutical interventions that were implemented in Denmark from February 2020 to March 2022.

2. In the abstract, “confined space” and “contact to known persons” should be separated. “Contact to known persons” should be reworded to “contact to infected individuals.”

Answer: Thank you for this. The abstract has been rephrased. Confined space is deleted and “Contact to known persons” is reworded to “contact to a person with known SARS-CoV-2 infection”, matching the wording performed through the entire manuscript. 

Abstract (Lines 20-23): Alcohol consumption and fitness centers were weakly associated with infection, in agreement with findings of our similar study conducted six month earlier. Apart from the two factors, no community activities were more common amongst cases, suggesting that community restrictions in place were adequate. The strongest risk factor for transmission was contact to an infected person. 

3. In the abstract, the sentence “For reference, we provide a timeline of non-pharmaceutical interventions in place in Denmark from February 2020 to March 2022” should be in the earlier part of the abstract to flow better.

Answer: Thank you for this suggestion, the sentence have been moved to an earlier part of the abstract (lines 12-13). The title of the manuscript have also been update, to include ‘overview of societal COVID-19 epidemic counter measures in Denmark’ for a better flow and understanding. 

4. In the abstract, the citation should not be included and should be removed.

Answer: The citation has been deleted. 

5. If there are any missing data, they should be elaborated on in the tables and provide information on how they were addressed. If not, please ignore this comment.

Answer: There was no missing data. In the analysis presented in table 3, all ‘do not know’ responses were excluded (stated in the footnote under the table). 

6. This is a suggestion, but authors should reconsider whether there is a need to do the analysis excluding cases who reported to be infected in their household (Table 4) when the authors are doing an analysis excluding individuals with close contact (Table 5)? I understand that the former data provided some factors that are statistically significant while the latter did not, but it seems arbitrary.

Answer: Thank you for this relevant suggestion. The purpose of Table 5 was to see if self-isolation could explain why more controls were often exposed than cases (resulting in OR estimates below 1). Persons who recently had been in close contact with an infected person would self-isolate, as recommended, and would therefore not be having activities in the community (if they had this information during their exposure period!). Removing those with known contact to an infected person did not change the results. 

The intension with Table 4, excluding household transmission, is to remove those exposed at home, i.e. those not relevant for assessing risk associated with activities in the community. 

We do not have detailed information about the setting of the close contact (those excluded in Table 5), and could thus be excluding transmission occurring during community activities if implementing your suggestion. No changes have been made to the manuscript.

7. The reason to exclude cases who are close contact is partly that cases most likely acquired the virus through this specific contact rather than other community exposures you are asking about in the questionnaire. This can also be mentioned in the manuscript.

Answer: Thank you this valid point. We unfortunately do not have detailed information about the setting of the close contact that would allow us to tease out which close contact occurred during community activities and which occurred at home or during non-community activity. We have mentioned this limitation in the manuscript.

Line 283-288 We were able to exclude cases exposed in their household, which allowed us to produce a more specific estimate to risk associated to activities in the community. Unfortunately we did not have detailed enough information about the setting of the close contacts not part of the household, to include only those occurring during activities in the community. This means that we may have included cases which may have been infected during home or during non-community activities, which may have weakened the associations.

Reviewer #2: The manuscript "Societal COVID-19 epidemic counter measures and activities associated with SARSCoV-2 infection in an adult unvaccinated population – a case-control study in Denmark, June 2021" is a valuable contribution. Although it does not providing ground-breaking or surprising results, it adds evidence to a steep learning curve from COVID-19 pandemic for epidemiologists, social scientists and policymakers. This contribution is a good example of robust and rapid evidence for policy that can be immediately applied during the crisis and build evidence to better prepare for future pandemics.

General suggestions on methods:

- Since the outcome of interest is tested positive for SARS-CoV-2, could the authors elaborate on the national criteria for testing that were used during the study period? It is relevant for the international audience, since in every country a "case" could have a slightly different probability of being identified. It is mentioned in the discussion that there was massive testing, possibly including asymptomatic people without any indications, in the occupational setting. In such a case, it is worth introducing this context, and addressing the issue of RT-PCR validity and reliability in the limitation section.

Answer: Thank you for this comment, additional information on the testing recommendations and references have been added to the discussion. 

Lines 294-301: In the study period, participation in societal activities generally required regular testing regardless of symptoms as part of the corona passport strategy in our unvaccinated, not previously infected study population. Persons who had had contact with a SARS-CoV-2 infected person or had symptoms of COVID-19 were recommended one or more tests and additionally recommendations or regular screening tests existed for certain professions [28, 29]. During the period around 11,000-26,000 tests were performed per 100,000 population [30] and seroprevalence studies based on blood donors showed that a low proportion the adult healthy population of Denmark had been infected [31]. Therefore, we believe that controls were unlikely to have been positive without knowing.

- Could the authors explain a bit more why they applied individual matching? With such massive exposures and such common infection, it would be perhaps better to use unmatched design and check for the effect of age and sex on the infection risk? The purpose of the control group is to represent exposures in the source population and matching is a way to prevent confounding. If age and sex are not confounders in a given scenario, matching can actually introduce confounding. I recommend adding to the Methods section a short explanation/rationale to apply individual matching.

Answer: Thank you for this question, we applied individual matching as we assume that the matching variables were confounders. We assume that age, sex and place of residence affect your behaviour (how often you are dinning at a restaurant, visiting bars, drinking alcohol, going to fitness, but also adherence to official guidelines), number of contacts and the local reproduction rate. No adjustment have been made to the manuscript. 

- Could the authors explain the rationale behind excluding residents previously vaccinated and previously tested positive by RT-PCR? We know now that previous infection or vaccination does not prevent new infections. If one wants to investigate the effect of community exposures on disease risk, maybe it would be a better idea to not exclude infections occuring >6 months previously, for example. Maybe the authors just assumed that with a new disease circulating for a short time, it is safe to assume that all were recently vaccinated and all infected to date were "recently" infected and thus were immune to SARS-CoV-2 infections... These assumptions should be however better explained and addressed in the limitations section.

Answer: Thank you for this comment. Michlmary et al. 2022, investigated the observed protection against SARS-CoV-2 reinfection following a primary infection. The study was performed in Denmark, in the same period as our study was performed. They showed that SARS-CoV-2 infection and vaccination offered a high level of protection against reinfection. Based on this, we assumed that the risk of infection/reinfection was not the same for those previously SARS-CoV-2 positive, not previous infected and those vaccinated against SARS-CoV-2. In order not to include people with different risk of infection, the study was performed among people who had not previously tested positive. The following adjustment have been made:

Lines 91-92: Only controls unvaccinated and not previously infected by 12 June 2021 were included to match the risk of infection profile of the cases.

Reference: Michlmayr, D., Hansen, C. H., Gubbels, S. M., Valentiner-Branth, P., Bager, P., Obel, N., Drewes, B., Møller, C. H., Møller, F. T., Legarth, R., Mølbak, K., & Ethelberg, S. (2022). Observed protection against SARS-CoV-2 reinfection following a primary infection: A Danish cohort study among unvaccinated using two years of nationwide PCR-test data. The Lancet regional health. Europe, 20, 100452. https://doi.org/10.1016/j.lanepe.2022.100452

- I recommend using the term matched odds ratios (mOR) instead of odds ratios, to reflect the methodological approach. It should be updated in the entire manuscript, including the abstract.

Answer: Thank you. We have now specified the type of odds ratios where the specification lacked. 

Lines 9-11: We determined matched odds ratios (mORs) and 95% confidence intervals (95%CIs) by conditional logistical regression with adjustment for household size and immigration status. 

Lines 15-19: Reporting close contact with an infected person who either had or did not have symptoms resulted in mORs of 20 (95%CI:9.8-39) and 8.5 (95%CI 4.5-16) respectively. In contrast, community exposures were generally not associated with disease; several exposures were negatively associated. Exceptions were: attending fitness centers, mOR=1.4 (95%CI:1.0-2.0) and consumption of alcohol in restaurants or cafés, aOR=2.3 (95%CI:1.3–4.2). 

- As mentioned in previous comments, the study has several limitations, not thoroughly revised by the authors. I recommend to be a bit more comprehensive in this matter!

Answer: Thank you for this recommendation. In order to accommodate, the section in the discussion covering the methodological strengths and limitation have been revised. The section now includes concerns about recall bias and parts have been rephrased to improve readability.

Lines 267-301: Methodological strengths and limitation outlined in our previous 2020-study also apply for the current study. Among the limitations of the first study was the small sample size, therefore we went from 600 to 1000 participants to strengthen the power of the present study. We did not find any high mOR with wide confidence intervals, indicating type I errors, and therefore we consider our sample size to have been sufficient to identify associations had they existed. Compared to our first study, we also shortened the exposure period inquired about, aiming to provide more specific estimates of associations. The use of the Danish Vaccination Registry enabled us to swiftly and objectively exclude those who had been vaccinated by the time of the study and used the Danish Microbiology Database to exclude those previously infected.

A potential bias would arise from systematic differences in behavior between cases and controls. Persons who recently had been in close contact with a person with SARS-CoV-2 infection, would, if they were aware of the exposure at the time, likely have been in self-isolation and would therefore not be active in the community. Because we for many activities found controls to be more exposed than cases (resulting in OR estimates below 1), we were suspicious of such a bias being at play. To explore this further, we performed a sensitivity analysis, in which we excluded all participants who reported to have been close contacts to infected persons. This did not change the results. We were able to exclude cases exposed in their household, which allowed us to produce a more specific estimate to risk associated to activities in the community. Unfortunately we did not have detailed enough information about the setting of the close contacts not part of the household, to include only those occurring during activities in the community. This means that we may have included cases which may have been infected at home or during non-community activities, which may have weakened the associations. Another potential systematic difference between cases and controls is their ability to recall activities in the period in question, if cases already during contact tracing had been asked to recall relevant activities, they may have been more likely to report such activities adequately, which could have influenced the results. Another potential concern relates to the selection of controls. We used matched controls sampled from the general population, which was made possible because of our access to the Danish Civil Registration System. A different approach, which we did not opt for, would have been control selection with recruitment from the pool of persons testing negative in PCR test in the same period as the pool of cases tested positive. This option has been used by others, and we cannot say how it would have influenced on our results [5, 9]. In the study period, participation in societal activities generally required regular testing regardless of symptoms as part of the corona passport strategy in our unvaccinated, not previously infected study population. Persons who had had contact with a SARS-CoV-2 infected person or had symptoms of COVID-19 were recommended one or more tests and additionally recommendations or regular screening tests existed for certain professions [28, 29]. During the period around 11,000-26,000 tests were performed per 100,000 population [30] and seroprevalence studies based on blood donors showed that a low proportion the adult healthy population of Denmark had been infected [31]. Therefore, we believe that controls were unlikely to have been positive without knowing.

Minor corrections:

- Abstract line 10: Please move the numbers to the results

Answer: The number of included cases and controls have been removed to the results. 

- Abstract lines 20-21: Do not include citations in the abstract.

Answer: The reference have been deleted from the abstract. 

- Introduction line 39: Why to refer to case-control studies? Do authors refer to studies in Denmark? It would be good to be more precise...

Answer: The sentence have been updated. 

Line 44: By use of a case-control design, researchers all over the world have aimed to identify determinants, private and societal, for SARS-CoV-2 infection.

- Methods line 70: typo in "ministries". 

Answer: Thank you for noticing this. It has been corrected. 

Line 75-76: The information was retrieved from relevant Danish government ministries and from the national COVID-19 communication partnership (coronasmitte.dk). 

- Results line 163: did the authors mean: "Compared to eligible cases, eligible controls [...]"?

Answer: Thank you for making us aware that this sentence might be misunderstood. We refer to the difference between eligible cases (unvaccinated individuals between 18–49 years old, with an address in Denmark, and an RT-PCR confirmed SARS-CoV-2 infection in the period from 8 to 12 June 2021) and included cases in the study. We have made minor adjustments to the manuscript to make the point clearer.

Lines 174-175: Compared to eligible cases diagnosed during the period, the recruited cases were less likely to have migrant background (data not shown).

- Discussion 279: "conducting" not "conduction"

Answer: thanks, the spelling mistake have been corrected. 

Lines 298-303: Since no determinants for infection were identified, we conclude that overall societal restrictions in use in the spring of 2021 were adequate to reducing any potential risk associated with community activities such as participating in cultural events, dining at restaurants, shopping and public transportation; exceptions being fitness centers and alcohol consumption. Instead, a major risk factor for transmission was contact to an infected person and transmission primarily took place via infected colleagues or family members.

Reviewer #3: This is a case-control study attempting to identify individual and contextual risk factors for SARS-CoV-2 infection in Denmark. The study is well written and has a sound methodology. However, I have a few questions that I believe should be addressed:

- In the abstract (line 4) the authors mention they aim at testing the efficiency of such measures - however efficiency requires a resources dimension - do the authors believe they have addressed this?

Answer: Thank you for this comment. We have revised the first part of the abstract and the conclusion so that it is more clear. 

Lines 2-4: Measures to restrict physical inter-personal contact in the community have been widely implemented during the COVID-19 pandemic. We studied determinants for infection with SARS-CoV-2 with the aim of informing future public health measures. 

Lines 20-23: Alcohol consumption and fitness centers were weakly associated with infection, in agreement with findings of our similar study conducted six month earlier. Apart from the two factors, no community activities were more common amongst cases, suggesting that community restrictions in place were adequate. The strongest risk factor for transmission was contact to an infected person.

- At the end of introduction (lines 60-61) authors mention the "present an overview of the official restrictions". This seems a bit misleading as the authors actually prepare one as part of their work. I suggest this to be rephrased.

Answer: The sentence have been rephrased. 

Lines 59-61: Here we present the results of a second national case-control study of risk factors for infection. For context and reference, we further created an overview of the official restrictions that have been in place in Denmark throughout the COVID-19 epidemic.

- Authors have excluded hospitalized and vaccinated patients. Can they comment on how this might have influenced the results and how this can affect results generalizability? 

Answer: .We studied determinants for infection with SARS-CoV-2 and were primarily interested in community exposures, and therefore we excluded cases and controls who had been hospitalized for more than 12 hours in the period of interest. 

Cases and controls who have been vaccinated before the 8 of June 2021 was also excluded. Michlmary et al. 2022, investigated the observed protection against SARS-CoV-2 reinfection following a primary infection. The study was performed in Denmark, in the same period as our study was performed. They showed that SARS-CoV-2 infection and vaccination offered a high level of protection against reinfection. Based on this, we assumed that the risk of infection was not the same for vaccinated and non-vaccinated persons. In order not to include people with different risk of infection, the study was performed among people who had not previously tested positive. That the study population were unvaccinated not previously infected persons limits the generalizability to the vaccinated – at the time older – population. Conclusions generalizable to the vaccinated population would have required a different design, for example with two or more case and control groups. No adjustment have been made to the manuscript. 

Reference: Michlmayr, D., Hansen, C. H., Gubbels, S. M., Valentiner-Branth, P., Bager, P., Obel, N., Drewes, B., Møller, C. H., Møller, F. T., Legarth, R., Mølbak, K., & Ethelberg, S. (2022). Observed protection against SARS-CoV-2 reinfection following a primary infection: A Danish cohort study among unvaccinated using two years of nationwide PCR-test data. The Lancet regional health. Europe, 20, 100452. doi:10.1016/j.lanepe.2022.100452

- Data collection: The exposure period ranged from eight to two days before the symptom onset (or test for asymptomatic) for cases. However I could not identify what was the index date for controls - what was the 6-day refer to? Can the authors clarify?

Answer: The 6-day period for controls, were the same as for their matched case. We have made minor adjustments to the text, and hope it is clearer now:

Lines 113-114: The 6-day period ran from eight to two days prior to symptom onset (or test date if asymptomatic) for cases and the same 6-day period for the matched control. 

- Some of the analysis reported in Table 1 and not described in the methods. I suggest this to be added.

Answer: Thank you for this suggestion, the methods section have been updated accordingly. 

Lines 139-141: We compared basic demographic characteristics (country of origin, household size, number of contacts and employment status) of cases and controls. We used Chi-Square to test for overall differences and matched logistic regression to test for intergroup differences.

- The authors emphasize the alcohol consumption in their results. However this has not been given an in-depth consideration in the discussion. Why the authors believe they have identified this results? Is it really related with the alcohol consumption or it this a proxy for other types of behaviours?

Answer: Thanks for this point. The following have been added to the discussion:

Lines 229-237: As also speculated previously [10], activities potentially involving heavy breathing, excretions of aerosols, close interaction and touching of multiple surfaces in a closed indoor environment could constitute an environment prone to SARS-CoV-2 transmission, compared to outdoor sport activities. In the same line is the association with alcohol when present at restaurants or cafés, where alcohol may be a proxy for reduced awareness of protective behavior and adherence to IPC recommendations, increasing the likelihood of closer interactions and with that SARS-CoV-2 transmission. On the other hand, many other investigated activities, notably use of public transportation, supermarkets and cultural and sports gatherings were as common in cases than amongst controls.

- There are negative results in for a series of contexts. Have the authors explored these results in more depth? They raise an hypothesis is the discussion (lines 261 onwards) which has not been corroborated by the sensitivity analysis. Do you see any alternative reasons for these results. Could remote work play a role here?

Answer: Recall bias have been added as a potential systematic bias between cases and controls. However, it has not been possible to investigate this potential bias. Unfortunately we do not have information on remote work. 

Lines 280-283: Another potential systematic difference between cases and controls is their ability to recall activities in the period in question, if cases already during contact tracing had been asked to recall relevant activities, they may have been more likely to report adequately, which can have influenced the results. 

- The authors claim their results support the "efficiency" of existing measures. However I am not entirely convinced here. In fact there is no counterfactual to assess the effectiveness (which I believe is what in fact authors intended to assess) or the impact of those measures. Please comment.

Answer: We have updated the main text and conclusion to make the message we wanted to convey clearer. We interpret our finding that we, with the exception of alcohol and fitness, did not identify any risk associated with community activities, as that the measures in place were adequate to reduce any potential risk to a level where we could not find a difference between cases and controls. Yes, transmission have happened in those settings, but not to an extent where we would be able to predict where such a transmission would be more likely to occur. Contact to an infected person was, not surprisingly, by far the strongest risk factor for infection. So is state that as we found no single determination to be associated with SARS-CoV-2 infection and because transmission appeared to primarily take place via contact to a person with known SARS-CoV-2 infection, community restrictions in place were efficient. 

Abstract (Lines 20-23): Alcohol consumption and fitness centers were weakly associated with infection, in agreement with findings of our similar study conducted six month earlier. Apart from the two factors, no community activities were more common amongst cases, suggesting that community restrictions in place were adequate. The strongest risk factor for transmission was contact to an infected person.

Conclusion (Lines 296-304): In conclusion, we show results of a study of risk factors for SARS-CoV-2 infection and compare with a similar study done six month earlier. Under the constraints of the methodology of case-control studies, no major determinants for infection were identified. Since no determinants for infection were identified, we conclude that overall societal restrictions in use in the spring of 2021 were adequate to reducing any potential risk associated with community activities such as participating in cultural events, dining at restaurants, shopping and public transportation; exceptions being fitness centers and alcohol consumption. Instead, a major risk factor for transmission was contact to an infected person and transmission primarily took place via infected colleagues or family members. Finally, we provide a timeline of non-pharmaceutical interventions that were implemented in Denmark from February 2020 to March 2022.

Reviewer #4: The authors aim to identify determinants of Sars-COV2 infection in the Denmark in June 2021 using a case –control study. They compare exposure to determinants among

1) 500 cases unvaccinated; 18-49 years old, residents in Denmark, with a RT-PCR test positive result between 8-12 June that were not hospitalized; had not travelled abroad

2) With 529 controls matched on year of birth, sex and municipality with the cases (1 to 1 match).

The 6 days (from Day -8 to Day -2 prior to symptom or test +) exposure to determinants was measures by telephone interviews between 15 and 24 June. Conditional logistic regression is correctly use for the analysis.

- The population from which cases and controls are coming from does not look identical concerning previous infection which is an exclusion criteria for controls but no clear info is provided about the cases. For the community transmission, given the restrictions in place, it means that we could have only controls with a recent negative test but we could have cases with a previous infection or a recent negative test who access public spaces. A subgroup analysis could be performed excluding cases with previous infection and their matched controls, to account for that.

Answer: Thank you for this comment. We apologise that it was not clearly described in the methods, but cases previously infected by 8 June 2021 were not included. Meaning that, cases and controls was similar concerning previous infection status. This have been added to the methods and adjusted in the abstract. 

Lines 5-6: We conducted a national matched case-control study among unvaccinated not previously infected adults aged 18-49 years

Lines 89-91: Eligible cases were unvaccinated individuals between 18–49 years old, not previously infected by 8 June 2021, with an address in Denmark, and an RT-PCR confirmed SARS-CoV-2 infection in the period from 8 to 12 June 2021.

- In order to understand the results without the need to consult other articles a more detailed description of the questionnaire used is necessary.

Answer: Thank you for this suggestion. A description of the included community exposures have been included. 

Lines 122-126: The community exposures inquired about were same as in our first study, and included activities as dining at restaurants, going to bars, shopping, participating in sport activities, and religious events or events involving singing etc. along with questions on if these activities took place indoors or outdoors, or involved consumption of alcohol. In contrast to the first study, we did not include questions on protective behavior and adherence with measures.

-The conclusion that Contact with another individual with a known infection as the main determinant for SARS-CoV-2 infection is based on sound methodology and nicely reflected in the different sections. Given the strength of the association even adjusting for other determinants will not modify the findings and or conclusions.

Sub-group analysis for the community transmission section.

The analysis for each determinant is performed separately adjusted for migration back ground, household size and the matching variables. There is no analysis performed by adjusting for all determinants, or for determinants associated with risk of infection in the bivariate analysis.

Answer: Thank you for this comment. As we do not identify any strong associations, data does not give rise to additional analyses. We fear that any multivariable (apart from the confounding factors adjusted for) analyses will strain our data to a point not meaningful. 

- Based on the results from table 4 some of the interpretations about differences between cases and controls are not fully correct. In the results section it states that cases are more likely than controls to have consumed alcohol in a bar. By looking at table 4, the CI95% of the OR for alcohol consumption in a Bar includes the 1 meaning that the difference is not statistically significant. The same logic applies for the indoor sport activities, indoor fitness center and outdoor sport activities. The difference between cases and controls is not statistically significant. The results and discussion sections are to be updated accordingly concerning the community transmission part. (no significant difference for indoor fitness centers, indoor sport activities, alcohol consumption in bars )

Answer: Thank you for this point. We have added that the trends we found, that more cases than controls consume alcohol at bar and indoor cultural events are not significant. 

Lines 183-188: Controls were more likely to report having been to restaurants than cases, mOR: 0.66 (95% CI: 0.49-0.90). However, cases were more likely than controls to report, that they or others in their company had consumed alcohol during the restaurant visit, adjusted odds ratio (aOR): 2.3 (95% CI: 1.3-4.2). The same trend was seen for bar and indoor cultural events, where more controls than cases reported to have visited a bar and participated in indoor cultural events, but more cases than controls reported alcohol consumption (non-significant) (Table 4). 

- It seems that the study is underpowered to detect community determinants of SARS-COV2 infection. Considering a matched 2 controls 1 case design could partially resolve that problem. In the multivariate model, only determinants associated with outcome in the bivariate analysis could be entered.

Answer: Thank you for this comment. We did not find any high mOR with wide confidence intervals (except within the sensitivity analysis), which could be a sign of a type I error, and we therefore believe the sample size would be sufficient to identify associations if they existed. Also, we had beforehand performed a power estimation; this is described in the MS. We have added a comment on this in the limitation section. 

We did not find several community exposures to be associated with SARS-CoV-2, and therefore we did not find a multivariate model relevant. 

Lines 268-270: We did not find any high mOR with wide confidence intervals, indicating type I errors, and therefore we consider out sample size to be sufficient to identify associations if they existed.

- Other

Page 4 line 56 (Introduction section) Please rephrase

Society was gradually reopening, with now only societal restrictions in place for those individuals who were vaccinated, had recovered from infection or recently tested negative

In contradiction with Line 71-72 (methods section)

Answer: Thank you for this comment. The introduction have been corrected. 

Lines 56-58: Society was gradually reopening, with now only societal restrictions in place for those individuals who were not vaccinated or protected from previous infection and for those who did not have a recently negative SARS-CoV-2 test.

- Table 1 Please provide OR for non matching variables. From the bivariate analysis it looks like the migration background and the number of contacts are associated with the risk of becoming a case.

Answer: Table 1 have been updated and now the table includes the p-value for the Chi-Square test for overall difference between cases and controls, as well the p-values for the matched odds ratios have been included. The methods section have also been updated

Lines 139-141: We compared basic demographic characteristics (country of origin, household size, number of contacts and employment status) of cases and controls. We used Chi-Square to test for overall differences and matched logistic regression to test for intergroup differences.

Table 1. Number and proportion of included cases and controls by demographic characteristics and p-value of test for deviations, Denmark, June 2021.

Demographic characteristics Included cases

(n=500), n (%) Matched controls

(n=529), n (%) 

P value1

Age group na

 18-24 years 199 (40) 212 (40) 

 25-34 years 159 (32) 170 (32) 

 35-44 years 85 (17) 88 (17) 

 45-49 years 57 (11) 59 (11) 

Sex na

 Male 263 (53) 283 (54) 

 Female 237 (47) 246 (47) 

Region na

 Capital Region of Denmark 195 (39) 200 (38) 

 Region Zealand 56 (11) 62 (12) 

 Region of Southern Denmark 61 (12) 66 (12) 

 Central Denmark Region 117 (23) 128 (24) 

 North Denmark Region 71 (14) 73 (14) 

Migration background2 <0.05

 Denmark 403 (81) 462 (87) Ref.

 Western country 26 (5.2) 23 (4.3) 0.34

 Non-western country 71 (14) 44 (8.3) <0.05

Household size3 0.64

 1 78 (16) 91 (17) Ref.

 2 141 (28) 165 (31) 0.92

 3 109 (22) 100 (19) 0.22

 4 105 (21) 108 (20) 0.46

 ≥5 67 (13) 65 (12) 0.39

Number of contacts <0.05

 0-5 290 (58) 208 (39) Ref.

 6-10 88 (18) 156 (29) <0.05

 11-15 47 (9.4) 51 (9.6) 0.06

 16-20 27 (5.4) 30 (5.7) 0.07

 21-49 30 (6.0) 52 (9.8) <0.05

 Over 50 18 (3.6) 32 (6.0) <0.05

Employment status 0.34 

 Employed 307 (61) 339 (64) Ref.

 Student 150 (30) 138 (26) 0.06

 Other 43 (8.6) 52 (9.8) 0.62

1 P-value for Chi-Square test for deviation between included cases and matched controls (italic). P-value for mOR (non-italic) 

2 Non-Danish migration background was defined as first or second generation immigrants from either Western countries (primarily neighboring European countries) or from non-Western countries (the five most frequent being Turkey, Iraq, Lebanon, Pakistan, and Somalia).

3 Number of registered persons on the same address.

In table 2 there is a total of 516 cases reported whereas in the results section only 500 cases are reported as included, please double-check or clarify

Answer: Thank you for noting this. 

In table 2, 500 cases were included and the 500 cases indicated 516 likely places of infection. The denominator in the table have been changed to 500 instead of 516. This is stated under the table. Additional, the 1, have been moved to the title, so make it more clear. 

Table 2. Number and proportion of likely place of infection as indicated by cases (n=500)1.

Likely place of infection n (%)

Household 105 (21)

Friends/other family than household 80 (16)

Workplace 84 (17)

Education 28 (5.6)

Leisure activities 27 (5.4)

Other events 13 (2.6)

Public transport 8 (1.6)

Other place/exposure 68 (14)

Don’t know 103 (21)

1It was possible to indicate more than one likely place of infection (516 answers included from 500 cases).

Table 3 Please provide adjusted OR for migration background and household size as well.

Answer: Thank you for this suggestion, we have added the OR for migration and household size from the type of contact model in table 3:

Table 3: 

Table 3. Number, proportion and matched odds ratios (mOR) related to type of contact with a person with known SARS-CoV-2 infection (with/without symptoms), Denmark, June 2021.

Type of contact with infected person with/without symptoms Cases (n=483), n (%) Controls (n=512), n (%) mOR (95% CI)

No contact with infected person 256 (53) 472 (92) Ref.

Other contact, without symptoms 10 (2.1) 6 (1.2) 3.04 (0.93-9.99)

Other contact, with symptoms 14 (2.9) 6 (1.2) 3.25 (1.15-9.19)

Close contact, without symptoms 73 (15) 16 (3.1) 8.53 (4.52-16.11)

Close contact, with symptoms 130 (27) 12 (2.3) 20 (9.80-39.49)

Migration background 

 Denmark 391 (81) 448 (88) Ref.

 Western country 25 (5.2) 23 (4.5) 1.13 (0.52-2.43)

 Non-western country 67 (14) 41 (8.0) 2.17 (1.28-3.68)

Household size 

 1 75 (16) 86 (17) Ref.

 2 135 (28) 159 (31) 0.93 (0.56-1.54)

 3 105 (22) 98 (19) 1.07 (0.63-1.79)

 4 102 (21) 106 (21) 1.25 (0.72-2.18)

 ≥5 66 (14) 63 (12) 0.88 (0.47-1.64)

Note mOR estimates were adjusted for type of contact with/without symptoms, migration background and household size. 

All ‘do not know’ responds were excluded in the analysis. 

General Feedback

Please standardize the reporting of OR and CI95% by using everywhere 2 decimals (1,01 instead of 1,0 or 39,00 instead of 39).

Inclusive tables with absolute numbers and proportions as well as OR(CI95%) of both bivariate and multivariate analysis would be more easy to read and interpret.

Answer: Thank you for this comment. We would be sad to report more than two significant digits on percentages and in the text, as we believe having only two significant digits improves readability and does not compromise the interpretation of the results. If the editor of PLOS ONE insist, we will of course adjust. 

Additional revisions: 

Cajar et al., 2022 have been published meanwhile, so the reference have been updated. 

Lines 255-259: Another Danish case-control study, performed in October to December 2020 found associations similar to those seen in our studies [5]. The most important risk factor identified was contact to an infected person. Moreover, this study also found an association with fitness centers but not with shopping, use of public transport and participating in outdoor sport activities. Contrary to our findings, participation in indoor sport activities, larger events, and restaurant and bar visits were identified as risk factors [5]

Reference: Cajar MD, Tan FCC, Boisen MK, Krog SM, Nolsoee R, Collatz Christensen H, et al. Behavioral factors associated with SARS-CoV-2 infection. Results from a web-based case-control survey in the Capital Region of Denmark. BMJ Open. 2022;12(6):e056393. doi: 10.1136/bmjopen-2021-056393

The link in reference [11] have been updated. 

Reference: The Danish Ministry of Health. Orientering om ændringer i regler vedr. coronapasset pr. 21. maj 2021. https://sum.dk/Media/637571404984193343/Orientering%20om%20%C3%A6ndringer%20i%20regler%20vedr.%20coronapas%20pr.%20215.pdf.

---

## [Decision Letter · Decision Letter 1]

21 Sep 2022

PONE-D-22-13314R1Case-control study of activities associated with SARS-CoV-2 infection in an adult unvaccinated population and overview of societal COVID-19 epidemic counter measures in DenmarkPLOS ONE

Dear Dr. Ethelberg,

Thank you for submitting your manuscript to PLOS ONE. After careful consideration, we feel that it has merit but does not fully meet PLOS ONE’s publication criteria as it currently stands. Therefore, we invite you to submit a revised version of the manuscript that addresses the points raised during the review process.

All the reviewers agreed that your revision is much improved. Two reviewers still have some minor requests for changes/clarifications.   

We look forward to receiving your revised manuscript.

Kind regards,

Joël Mossong, PhD

Academic Editor

PLOS ONE

Journal Requirements:

Reviewers' comments:

Reviewer's Responses to Questions

**Comments to the Author**

1. If the authors have adequately addressed your comments raised in a previous round of review and you feel that this manuscript is now acceptable for publication, you may indicate that here to bypass the “Comments to the Author” section, enter your conflict of interest statement in the “Confidential to Editor” section, and submit your "Accept" recommendation.

Reviewer #1: (No Response)

Reviewer #2: All comments have been addressed

Reviewer #3: All comments have been addressed

Reviewer #4: (No Response)

2. Is the manuscript technically sound, and do the data support the conclusions?

Reviewer #1: Partly

Reviewer #2: Yes

Reviewer #3: Yes

Reviewer #4: Partly

3. Has the statistical analysis been performed appropriately and rigorously? 

Reviewer #1: Yes

Reviewer #2: Yes

Reviewer #3: Yes

Reviewer #4: Yes

4. Have the authors made all data underlying the findings in their manuscript fully available?

Reviewer #1: No

Reviewer #2: Yes

Reviewer #3: No

Reviewer #4: No

5. Is the manuscript presented in an intelligible fashion and written in standard English?

Reviewer #1: Yes

Reviewer #2: Yes

Reviewer #3: Yes

Reviewer #4: Yes

6. Review Comments to the Author

Reviewer #1: The authors addressed most of my comments in a sufficient manner. However, I am still not convinced about the validity of their conclusion, which is critical for this paper. As I mentioned before, even if the community restrictions in place were efficient, behaviors that are truly high-risk would have been identified as such in the present study design. For example, individuals who were positive would be more likely to have history of going to bars and restaurant (resulting in higher odds) if this specific activity is indeed a high-risk behavior in the context of the study population. This should be the case regardless of what type of community restriction is in place. Conversely, as observed in this study, if we do not see any association between a specific behavior and infection, then it would not be possible to conclude that “community restrictions in place were efficient” as authors have done. If history of going to bars and restaurant is really high-risk, then we would see higher ORs. If the policy implemented includes restricting the opening hours at bars and restaurants and if we see that there is a good association between going to bars and restaurants, then we can infer that the policy is appropriately targeting high-risk individuals and high-risk behaviors. Below paper illustrated this exact point: Behavioral factors associated with SARS-CoV-2 infection in Japan. Influenza Other Respir Viruses. 2022;16(5):952-961.

Reviewer #2: The authors have carefully considered all comments of reviewers. Therefore, I recommend to accept in the current form.

Reviewer #3: Many thanks for fully considering the comments submitted and having provided satisfactory replies. While my comments were fully addressed I have 2 questions from replies from other reviewers:

- Authors have mentioned type I errors while refering to issues with power (R#2 and R#4). However I believe they intended to refer to type II erros (failure to reject a null hypothesis when it is false). Please review.

- Authors have mentioned in response to R#2 - "During the period around 11,000-26,000 tests were performed per 100,000 population [30]". I suggest to clarify whether these include only PCR or also Rapid Antigen Tests.

Reviewer #4: The majority of the comments have been addressed. However there are still few remaining issues to be addressed

General Comment :

Please consistently report two decimals for OR CI95% throughout the text.

In the Results Section /

Community determinants of SARS-CoV-2 infection

Lines 212-214

Apart from this,controls were more likely to participate in indoor sport activities, mOR: 0.57 (95% CI: 0.32-1.0), and outdoor sport activities, mOR: 0.71 (95% CI: 2130.49-1.0).

Comment :

Please consistently report two decimals for OR CI95% throughout the text.

In the table 4 for in indoor sport activities , mOR: 0.57 (95% CI: 0.32-1.01), and for outdoor sport activities

mOR: 0.71 (95% CI: 0.49-1.03) does not support the conclusion that controls were more likely to participate in indoor sport and outdoor sport activities.

Discussion :

Lines 247-252

The pattern of risk factors seen in the current study was remarkably similar to what we found in our previous study, where also, apart from contact to infected persons with known infections, fitness centers and alcohol consumption in bars and in addition participation in events which involved singing, were identified as being associated with SARS-CoV-2 infection. This second study may therefore be seen as corroborating the findings of the first and it would appear that besides direct contact with infected individuals, under the restrictive measures in place in Denmark, use of fitness centers and social activity involving alcohol constituted actual risk factors.

Comment : Use of fitness centers and participation in events which involved singing are not significantly related to the risk of infection in the second study. Please rephrase.

Lines 334 -338

We therefore conclude that overall societal restrictions in use in the spring of 2021 were in fact adequate to reducing any potential risk associated with community activities such as participating in cultural events, dining at restaurants, shoppings and public transportation; exceptions being fitness centers and alcohol consumption. which did constitute risks.

Comment : Use of fitness center is not significantly related to the risk of infection in the second study. Please rephrase.

7. PLOS authors have the option to publish the peer review history of their article (what does this mean?). If published, this will include your full peer review and any attached files.

Reviewer #1: No

Reviewer #2: No

Reviewer #3: **Yes: **Andreia Leite

Reviewer #4: No

---

## [Author Response · Author response to Decision Letter 1]

20 Oct 2022

Rebuttal letter, 'Response to Reviewers'. Revision no 2.

PONE-D-22-13314R1

Case-control study of activities associated with SARS-CoV-2 infection in an adult unvaccinated population and overview of societal COVID-19 epidemic counter measures in Denmark

Please note: Changes to the reference list are mentioned at the end of this doc.

****

Reviewer #1: The authors addressed most of my comments in a sufficient manner. However, I am still not convinced about the validity of their conclusion, which is critical for this paper. As I mentioned before, even if the community restrictions in place were efficient, behaviors that are truly high-risk would have been identified as such in the present study design. For example, individuals who were positive would be more likely to have history of going to bars and restaurant (resulting in higher odds) if this specific activity is indeed a high-risk behavior in the context of the study population. This should be the case regardless of what type of community restriction is in place. Conversely, as observed in this study, if we do not see any association between a specific behavior and infection, then it would not be possible to conclude that “community restrictions in place were efficient” as authors have done. If history of going to bars and restaurant is really high-risk, then we would see higher ORs. If the policy implemented includes restricting the opening hours at bars and restaurants and if we see that there is a good association between going to bars and restaurants, then we can infer that the policy is appropriately targeting high-risk individuals and high-risk behaviors. Below paper illustrated this exact point: Behavioral factors associated with SARS-CoV-2 infection in Japan. Influenza Other Respir Viruses. 2022;16(5):952-961.

Author reply: Thank you for raising this important point. We only partly agree. The study was done while restrictions concerning eg restaurants and bars were in place. They for instance involved restricted opening times, restrictions in the number of patrons present and use of masks. It is possible that we’d have seen restaurants appear as risk factors in a (hypothetical) un-restricted scenario but that the risk was lowered by the restrictions in place, resulting in no excess risk (measured as an odds ratio here) being identified. It is true of course that we cannot know this. It is also possible that the restrictions were redundant or simply had no effect and that we’d have obtained identical results had they not been in place. For this reasons we previously used to word ‘suggesting’. 

However, we would agree that we might have stretched our conclusion too far and have therefore now changed the wording in the abstract and in the last conclusion paragraph of the MS. We now no longer say that the restrictions may have been adequate, rather that we cannot know this.

In the Abstract, the lines “Apart from these two factors, no community activities were more common amongst cases, suggesting that community restrictions in place were adequate.”

Have now been changed to: “Apart from these two factors, no community activities were more common amongst cases under the community restrictions in place during the study” (lines 18-19).

In the Conclusion, the revised text now reads:

“We conclude that transmission in the general community was of little importance, while instead, the major risk factor for transmission was contact to a known infected person and that transmission primarily took place via infected colleagues or family members. Our study could not directly measure the effect of the societal restrictions in place but it is not unreasonable to expect that these had an effect in reducing any potential risks associated with community activities such as participating in cultural events, dining at restaurants, shopping and public transportation.” (lines 307-312). 

**

Reviewer #2: The authors have carefully considered all comments of reviewers. Therefore, I recommend to accept in the current form.

Author reply: Thank you. No changes made to the manuscript.

**

Reviewer #3: Many thanks for fully considering the comments submitted and having provided satisfactory replies. While my comments were fully addressed I have 2 questions from replies from other reviewers:

- Authors have mentioned type I errors while refering to issues with power (R#2 and R#4). However I believe they intended to refer to type II erros (failure to reject a null hypothesis when it is false). Please review.

Author reply: Thank for you noticing this mistake. We have now rephrased the section:

Line 273: We did not find any high mOR with wide confidence intervals, which could indicating risk of type II errors, and therefore we consider our sample size to have been sufficient to identify associations had they existed

- Authors have mentioned in response to R#2 - "During the period around 11,000-26,000 tests were performed per 100,000 population [30]". I suggest to clarify whether these include only PCR or also Rapid Antigen Tests.

Author reply: Thank you for this comment. The number refers to PCR tests. This have now been added to the manuscript. 

Lines 300-301: During the period around 11,000-26,000 RT-PCR tests were performed per 100,000 population

**

Reviewer #4: The majority of the comments have been addressed. However there are still few remaining issues to be addressed

- General Comment :

Please consistently report two decimals for OR CI95% throughout the text.

Author reply: Thanks for this. We’ve tried to consistently report two significant figures for OR’s and CI’s. Reporting two decimals would not be correct as it would imply precision in measurements that is not present. We for instance report “mOR: 20 (95% CI: 9.8-40)”. Here the number of significant figures are two for all three reported numbers. It would not have been correct to report, eg: “mOR: 19.89 (95% CI: 9.81-40.01), or [as a more extreme example]: “mOR: 163,12 (95% CI: 9.81-1936.23)”. (Please, if interested, see also: https://en.wikipedia.org/wiki/Significant_figures). We note that reporting of decimals is a matter of precision but also journal style and that the journal will decide given acceptance of the MS.

- In the table 4 for in indoor sport activities, mOR: 0.57 (95% CI: 0.32-1.01), and for outdoor sport activities mOR: 0.71 (95% CI: 0.49-1.03) does not support the conclusion that controls were more likely to participate in indoor sport and outdoor sport activities.

Author reply: Thank you for this comment. We have now have been rephrased the following four sections: 

Lines 17-19: “Consumption of alcohol in restaurants or cafés, aOR=2.3 (95%CI:1.3–4.2) and possibly attending fitness centers, mOR=1.4 (95%CI:1.0-2.0) were weakly associated with SARS-CoV-2 infection, in agreement with findings of our similar study conducted six month earlier. Apart from these two factors, no community activities were more common amongst cases, suggesting that community restrictions in place were adequate. The strongest risk factor for transmission was contact to an infected person.”

Lines 192-195: “In total, 24% of cases and 19% of controls had used a fitness centre at least once during the 6-day exposure period, mOR: 1.40 (95% CI: 0.98-2.0), this trend was not seen for indoor sport activities, mOR: 0.57 (95% CI: 0.32-1.0), nor for outdoor sport activities, mOR: 0.71 (95% CI: 0.49-1.0).”

Lines 232-238: “This second study may therefore be seen as corroborating the findings of the first and it would appear that besides direct contact with infected individuals, under the restrictive measures in place in Denmark, social activity involving alcohol and possibly the use of fitness centers constituted actual risk factors. As also speculated previously [10], activities potentially involving heavy breathing, excretions of aerosols, close interaction and touching of multiple surfaces in a closed indoor environment could constitute an environment prone to SARS-CoV-2 transmission, compared to outdoor sport activities.”

Lines 309-314: “We therefore conclude that overall societal restrictions in use in the spring of 2021 were adequate to reducing any potential risk associated with community activities such as participating in cultural events, dining at restaurants, shopping and public transportation; exceptions being alcohol consumption and possibly use of fitness centers. Instead, a major risk factor for transmission was contact to an infected person and transmission primarily took place via infected colleagues or family members.”

- Discussion:

Lines 247-252

The pattern of risk factors seen in the current study was remarkably similar to what we found in our previous study, where also, apart from contact to infected persons with known infections, fitness centers and alcohol consumption in bars and in addition participation in events which involved singing, were identified as being associated with SARS-CoV-2 infection. This second study may therefore be seen as corroborating the findings of the first and it would appear that besides direct contact with infected individuals, under the restrictive measures in place in Denmark, use of fitness centers and social activity involving alcohol constituted actual risk factors.

Comment: Use of fitness centers and participation in events which involved singing are not significantly related to the risk of infection in the second study. Please rephrase.

Author reply: Thank you for this comment. We have modified the text accordingly: 

Lines 228-235: “The pattern of risk factors seen in the current study was remarkably similar to what we found in our previous study, where also, apart from contact to infected persons, fitness centers and alcohol consumption in bars and in addition participation in events which involved singing, were identified as being associated with SARS-CoV-2 infection. This second study may therefore be seen as corroborating the findings of the first and it would appear that besides direct contact with infected individuals, under the restrictive measures in place in Denmark, social activity involving alcohol and possibly the use of fitness centers constituted actual risk factors.”

- Lines 334 -338

We therefore conclude that overall societal restrictions in use in the spring of 2021 were in fact adequate to reducing any potential risk associated with community activities such as participating in cultural events, dining at restaurants, shoppings and public transportation; exceptions being fitness centers and alcohol consumption. which did constitute risks.

Comment: Use of fitness center is not significantly related to the risk of infection in the second study. Please rephrase.

Author reply: Thank you for this comment. We have now modified the text accordingly: 

Lines 309-312: “We therefore conclude that overall societal restrictions in use in the spring of 2021 were adequate to reducing any potential risk associated with community activities such as participating in cultural events, dining at restaurants, shopping and public transportation; exceptions being alcohol consumption and possibly use of fitness centers.”

*****

Author: This concludes our responses to the reviewer comments. Please not that in addition we have also made changes, as per Journal Requirements, to the reference list:

References:

The following changes were made to the reference list:

Ref. 12: The link in the reference list have been updated, as the original link wasn’t online anymore. 

The updated reference is: 

The Danish Ministry of Health Ministry of Industry Business and Financial Affairs and The Ministry of Employment. Retningslinjer om ansvarlig indretning af liberale serviceerhverv og køreskoler i lyset af udbruddet af COVID-19. https://em.dk/media/14213/retningslinjer-for-liberale-serviceerhverv-af-14-juni.pdf

Ref. 20: The link in the reference list have been updated, as the original link wasn’t online anymore.

The updated reference is: 

Aftale om yderligere genåbning pr. 21 maj 2021

https://coronasmitte.dk/nyt-fra-myndighederne/pressemoeder/aftale-om-yderligere-genaabning

Ref 23: The link in the reference has been deleted. 

Ref 30: The link to Statens Serum Instituts dashboard has been updated.

---

## [Editor Report · Decision Letter 2]

25 Oct 2022

Case-control study of activities associated with SARS-CoV-2 infection in an adult unvaccinated population and overview of societal COVID-19 epidemic counter measures in Denmark

PONE-D-22-13314R2

Dear Dr. Ethelberg,

We’re pleased to inform you that your manuscript has been judged scientifically suitable for publication and will be formally accepted for publication once it meets all outstanding technical requirements.

Kind regards,

Joël Mossong, PhD

Academic Editor

PLOS ONE
---

## [Editor Report · Acceptance letter]

8 Nov 2022

PONE-D-22-13314R2 

Case-control study of activities associated with SARS-CoV-2 infection in an adult unvaccinated population and overview of societal COVID-19 epidemic counter measures in Denmark 

Dear Dr. Ethelberg:

I'm pleased to inform you that your manuscript has been deemed suitable for publication in PLOS ONE. Congratulations! Your manuscript is now with our production department. 

Kind regards, 

on behalf of

Dr. Joël Mossong 

Academic Editor

PLOS ONE